# Ferroptotic cell death triggered by conjugated linolenic acids is mediated by ACSL1

Alexander Beatty [1✉], Tanu Singh[1], Yulia Y. Tyurina[2,3], Vladimir A. Tyurin[2,3], Svetlana Samovich[2,3], Emmanuelle Nicolas[1], Kristen Maslar [4], Yan Zhou[1], Kathy Q. Cai[1], Yinfei Tan[1], Sebastian Doll[5], Marcus Conrad [5,6], Aravind Subramanian[7], Hülya Bayır [2,3,8], Valerian E. Kagan [2,3,9,10,11,12], Ulrike Rennefahrt[13] & Jeffrey R. Peterson [1✉]

Ferroptosis is associated with lipid hydroperoxides generated by the oxidation of poly-unsaturated acyl chains. Lipid hydroperoxides are reduced by glutathione peroxidase 4 (GPX4) and GPX4 inhibitors induce ferroptosis. However, the therapeutic potential of triggering ferroptosis in cancer cells with polyunsaturated fatty acids is unknown. Here, we identify conjugated linoleates including α-eleostearic acid (αESA) as ferroptosis inducers. αESA does not alter GPX4 activity but is incorporated into cellular lipids and promotes lipid peroxidation and cell death in diverse cancer cell types. αESA-triggered death is mediated by acyl-CoA synthetase long-chain isoform 1, which promotes αESA incorporation into neutral lipids including triacylglycerols. Interfering with triacylglycerol biosynthesis suppresses ferroptosis triggered by αESA but not by GPX4 inhibition. Oral administration of tung oil, naturally rich in αESA, to mice limits tumor growth and metastasis with transcriptional changes consistent with ferroptosis. Overall, these findings illuminate a potential approach to ferroptosis, complementary to GPX4 inhibition.

[1] Fox Chase Cancer Center, Philadelphia, Pennsylvania, USA. [2] Department of Environmental and Occupational Health, University of Pittsburgh, Pittsburgh, PA, USA. [3] Center for Free Radical and Antioxidant Health, University of Pittsburgh, Pittsburgh, PA, USA. [4] Department of Biochemistry and Molecular Biology, Drexel University College of Medicine, Philadelphia, PA, USA. [5] Institute of Metabolism and Cell Death, Helmholtz Zentrum München, Neuherberg, Germany. [6] National Research Medical University, Laboratory of Experimental Oncology, Ostrovityanova 1, Moscow 117997, Russia. [7] Broad Institute of Harvard and MIT, Cambridge, MA, USA. [8] Department of Critical Care Medicine, Safar Center for Resuscitation Research, University of Pittsburgh, Pittsburgh, PA, USA. [9] Department of Chemistry, University of Pittsburgh, Pittsburgh, PA, USA. [10] Department of Pharmacology and Chemical Biology, University of Pittsburgh, Pittsburgh, PA, USA. [11] Department of Radiation Oncology, University of Pittsburgh, Pittsburgh, PA, USA. [12] Laboratory of Navigational Redox Lipidomics, IM Sechenov Moscow State Medical University, Moscow, Russia. [13] Metanomics Health GmbH, Berlin, Germany. ✉email: Alexander.Beatty@fccc.edu; Jeffrey.Peterson@fccc.edu

Ferroptosis is a form of non-apoptotic, iron-dependent, regulated cell death linked to lipid hydroperoxides, a form of reactive oxygen species (ROS)[1–7]. Interest in the therapeutic potential of inducing ferroptosis in cancer cells has led to a major research focus on inhibiting cellular antioxidant pathways that oppose ferroptosis. GPX4 is the only peroxidase known to efficiently reduce esterified, hydroperoxy fatty acids into unreactive alcohols[8]. Recently, ferroptosis suppressor protein 1 (FSP1) was identified as a second ferroptosis suppression mechanism through its recycling of coenzyme $Q_{10}$, a radical-trapping antioxidant[9,10]. Small molecules that trigger ferroptosis identified thus far include agents that inhibit GPX4 directly, molecules that deplete the GPX4 cofactor glutathione, and compounds that oxidize iron[1,9,11,12]. Lipid hydroperoxides are generated by non-enzymatic oxidation or by lipoxygenases or cytochrome P450 oxidoreductase[13–15]. Hydroperoxides disrupt membrane architecture, produce reactive aldehydes, and drive cell death[16]. In contrast to therapeutic targeting of antioxidant pathways, we have pursued a complementary strategy for ferroptosis induction based on enhancing the production of lipid hydroperoxides.

Hydroperoxides can spread by a free radical-mediated chain reaction leading to oxidation of adjacent polyunsaturated acyl chains, and consequently higher levels of this species increase vulnerability to peroxide propagation. Indeed, supplementing cells with arachidonic acid sensitizes them to ferroptosis triggered by GPX4 inhibitors[17,18]. Similarly, clear-cell carcinomas are rendered vulnerable to GPX4 inhibitors due to HIF-2α-dependent polyunsaturated lipid accumulation[19]. Conversely, increasing the relative proportion of monounsaturated acyl chains at the expense of polyunsaturated lipids suppresses sensitivity to ferroptosis[20,21]. Thus, increasing polyunsaturated lipids in cancer cells could enhance their vulnerability to ferroptosis. Importantly, this approach could exploit the propensity of cancer cells to scavenge fatty acids from their environment[22].

Here, we report that linolenic fatty acids with conjugated double bonds, produced by certain plant species, can induce ferroptosis in diverse cancer cells as a single agent. Conjugated linoleate triggers ferroptosis by a mechanism distinct from canonical ferroptosis inducers. Cell death is mediated by acyl-CoA synthetase long-chain family member 1 (ACSL1), implicating this isoform in ferroptosis. Furthermore, oral administration of tung tree seed oil, naturally rich in the conjugated linoleate α-eleostearic acid (αESA), inhibits tumorigenesis and metastasis in a murine breast cancer xenograft model. αESA metabolites are detected in tumors of treated mice and are associated with the expression of a ferroptotic gene signature. These results introduce a distinct class of ferroptosis inducers and offer insights into the molecular basis of ferroptotic sensitivity. The tractability of these dietary, pro-ferroptotic fatty acids addresses the current lack of effective GPX4 inhibitors for use in vivo and suggests an opportunity to exploit a metabolic liability across cancer subtypes.

## Results

### Glutathione depletion triggers ferroptosis in triple-negative breast cancer cells. 
Several triple-negative breast cancer cell lines are killed by the glutathione biosynthesis inhibitor buthionine sulfoximine (BSO) (Fig. 1a and ref. [23]). However, the specific cellular pool of ROS responsible for cell death is unclear. Glutathione depletion in some contexts induces ferroptosis[24], which is operationally defined by three criteria; its association with lipid peroxidation, the ability to suppress cell death with lipophilic antioxidants like ferrostatin-1 (fer-1), and iron-dependence[1]. Consistent with ferroptosis, cytotoxicity in BSO-treated BT-549 cells was blocked by fer-1 but not by Z-VAD-FMK, an inhibitor of apoptosis (Fig. 1a, b). Similarly, iron chelators deferoxamine

and deferiprone suppressed BSO-mediated cell death (Fig. 1c, Supplementary Fig. 1a). Furthermore, BSO treatment resulted in the accumulation of lipid peroxidation products and this was suppressed by fer-1 (Fig. 1d, e). Thus, glutathione depletion triggers death consistent with ferroptosis in BT-549 cells and identifies lipid hydroperoxides as the lethal ROS species underlying glutathione addiction in these cells.

### Sensitivity to ferroptosis is associated with the accumulation of polyunsaturated lipids.
Fer-1 suppressed BSO toxicity in half of the triple-negative breast cancer (TNBC) cell lines tested, consistent with ferroptosis (Fig. 1f, red bars). These cell lines were also generally more sensitive to ML162, a small molecule inhibitor of GPX4[25] (Welch's t-test, $p = 0.01$, Fig. 1g, Supplementary Fig. 1b). Thus, fer-1-suppressible cell death is a common response to glutathione depletion in a subset of TNBC cells and is associated with a greater dependence on GPX4 activity. This vulnerability was not due to decreased GPX4 expression (Supplementary Fig. 1c). We used ANOVA of our prior metabolomic data for these cell lines[23] to compare levels of individual metabolites between TNBC cell lines in which fer-1 rescued BSO-mediated cell death and those in which it did not (Fig. 1f), and found that two of the three most significantly differentially accumulated metabolites were linoleate (18:2)-substituted phosphatidylcholines (Fig. 1h, i). These membrane phospholipids were more enriched in cell lines that undergo fer-1-suppressible death. This finding suggested the hypothesis that polyunsaturated lipid enrichment may increase dependence on GPX4 and vulnerability to ferroptosis (Fig. 1g).

### αESA triggers cancer-selective ferroptosis.
We postulated that elevating polyunsaturated lipid levels might sensitize TNBC cells to ferroptosis. We tested a variety of fatty acids and identified α-eleostearic acid [(9Z,11E,13E)-octadeca-9,11,13-trienoic acid; αESA] (Fig. 2a) as a fatty acid that enhanced cell death in combination with BSO (Fig. 2b). αESA is abundant in certain plants (e.g. bitter melon and tung tree)[26] and was previously shown to suppress the growth of breast cancer cells in a manner reversible by antioxidants[27]. Others have suggested that αESA triggers apoptotic cell death[28–33] although a role for ferroptosis was not investigated. Unexpectedly, we found that αESA triggered cell death as a single agent and this death was suppressed by fer-1 (Fig. 2b), the iron chelator deferoxamine (Fig. 2c), and by the lipophilic antioxidant vitamin E (Supplementary Fig. 2a), and was associated with an increase in lipid peroxidation products that could be suppressed by fer-1 (Fig. 2d). Neither Z-VAD-FMK nor the necroptosis inhibitor nec-1s blocked αESA-induced cell death (Supplementary Fig. 2b, c). Together, these findings show that αESA induces ferroptosis as a single agent and distinguishes αESA from arachidonic and adrenic fatty acids, which sensitize cells to GPX4 inhibition but do not induce ferroptosis[13,17].

The non-transformed MCF-10A (Fig. 2e) and MCF-12A (Supplementary Fig. 2d) breast epithelial cell lines were resistant to death by αESA. By contrast, all TNBC cell lines were susceptible and, in all cases, death was prevented by fer-1 (Fig. 2b, f). Notably, even cell lines that did not undergo ferroptosis in response to glutathione depletion (Fig. 1f) were sensitive to αESA, pointing to a distinct mechanism of ferroptosis induction. Time-lapse microscopy of αESA-treated BT549 cells (Supplementary Movie 1) showed that during the latter part of 24 h of treatment, otherwise healthy appearing cells undergo a sudden death, morphologically similar to cell death by the canonical ferroptosis inducer ML162 (Supplementary Movie 2) and distinct from apoptosis (Supplementary Movie 3).

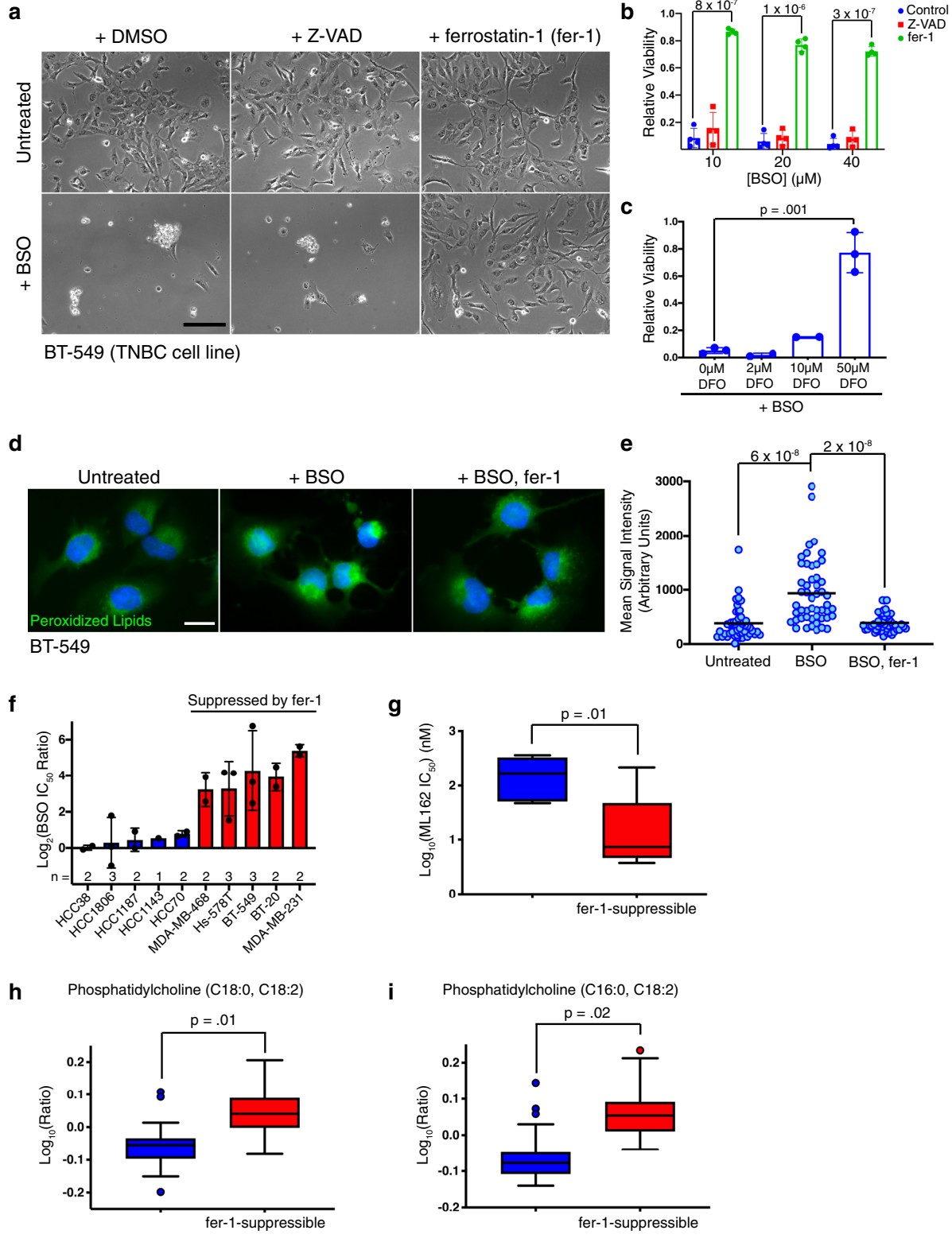

**αESA is incorporated into diverse cellular lipids**. To examine how αESA is metabolized, we conducted a lipidomic analysis of MDA-MB-231 cells treated with αESA for three hours, prior to overt cell death. Lipids containing acyl chains with 18 carbons and three double bonds (18:3), consistent with αESA, were rare in untreated cells (Fig. 2g, blue bars), but were found in diverse lipids in αESA-treated cells, including phospholipids and neutral lipids (green bars; Supplementary Data 1). Fer-1 co-treatment did

not dramatically alter the spectrum of lipid classes that incorporated 18:3 acyl chains (compare green & purple bars). The relative abundance of di- and triacylglycerol lipids increased ~2-fold in αESA and αESA + fer-1-treated cells compared to untreated controls, consistent with αESA being incorporated into storage lipids (Fig. 2h). αESA-induced lipidomic changes were not downstream consequences of ferroptosis because they were not replicated by ML162 treatment (orange bars in Fig. 2g, h).

**Fig. 1 Glutathione depletion triggers ferroptosis in a subset of TNBC cell lines and elevated polyunsaturated fatty acid levels are associated with vulnerability to ferroptosis. a** Light micrographs of BT-549 cells treated with 10 μM BSO for 72 h in the presence or absence of 20 μM Z-VAD-FMK or fer-1. Scale bar represents 200 μm. **b** BT-549 relative cell viability after 72 h of BSO treatment with vehicle, 20 μM Z-VAD-FMK, or 2 μM fer-1 ($n = 4$ independent experiments). Error bars, here and below, denote standard deviation centered on the mean. The numbers above the brackets are p values from Student's t-tests (two-sided) unless otherwise noted. p values were not corrected for multiple testing unless stated. **c** Relative viability of BT-549 cells treated with 10 μM BSO and the indicated concentration of deferoxamine (DFO) for 72 h ($n = 3$). Values were normalized to account for loss of viability associated with DFO. **d** Representative fluorescent micrographs of BT-549 cells treated with vehicle, BSO (50 μM), or BSO and fer-1 (2 μM) for 48 h. Green corresponds to cellular macromolecules modified with peroxidized lipid breakdown products (Click-iT lipid peroxidation detection kit, ThermoFisher Scientific). DNA is stained blue. Scale bar represents 20 μm. **e** Quantitation of lipid peroxidation products from individual cells ($n = 50$ per condition) in **d**. Lines represent the mean. **f** The $\log_2$-transformed ratio of the $IC_{50}$ of BSO in the presence or absence of 2 μM fer-1 for each cell line. Cell lines in which fer-1 reduced cell death from BSO > 8-fold are designated "fer-1-suppressible". **g** Box and whiskers plot of the $\log_2$-transformed $IC_{50}$ values for ML162 in non-fer-1-suppressible (blue) and fer-1-suppressible (red) TNBC cell lines (as defined in **f**). The line represents the median value, the box defines the interquartile range (25th to 75th percentile), and the whiskers show minimum and maximum values. **h** Box plots showing the $\log_{10}$-transformed, median-normalized relative levels of phosphatidylcholine (C18:0, C18:2) and **i** phosphatidylcholine (C16:0, C18:2) in non-fer-1-suppressible and fer-1-suppressible TNBC cell lines. Each cell line is represented by at least 5 replicates. p values based on ANOVA and corrected for multiple testing. The line shows the median value, the box shows the interquartile range, the whiskers represent the upper and lower adjacent values, and outliers are shown as dots. Source data are provided as a Source Data file.

**Structure-activity analysis of conjugated PUFAs**. To characterize structural features of αESA required for ferroptosis, we screened related conjugated and unconjugated polyunsaturated fatty acids for death induction. αESA possesses conjugated double bonds at carbons 9, 11, and 13 with cis, trans, trans stereochemistry, respectively (Fig. 2a). We examined cis-trans stereoisomers of αESA and found that isomerization at position 9 (β−eleostearic acid) or at both 9 and 13 (catalpic acid) retained cell death activity while isomerization at position 13 alone (punicic acid) led to a ~4-fold reduction in potency (Fig. 3a). Jacaric acid (conjugated 18:3, cis-8, trans-10, cis-12) showed the most potent cell killing (1.8 μM $IC_{50}$), demonstrating that shifting double bond positioning while maintaining the sequential cis, trans, cis stereochemistry of punicic acid, did not disrupt activity. α−calendic acid (conjugated 18:3, trans-8, trans-10, cis-12), a stereoisomer of jacaric acid, was similarly potent. Thus, cell killing is not strictly dependent on the precise positioning or stereochemistry of double bonds within the aliphatic chain. Conjugated linoleic acid (18:2), which shares similar position and stereochemistry of its two double bonds with αESA, was a poor inducer of cell death, however, suggesting that the third conjugated double bond is required for activity. Furthermore, all tested non-conjugated polyunsaturated fatty acids (PUFAs) were much less potent single-agent inducers of cell death (Supplementary Fig. 2e) including arachidonate, which enhances ferroptosis in some contexts[17,18]. Cell killing by arachidonate could not be rescued by fer-1, liproxstatin or deferoxamine, highlighting an important distinction between the activity of conjugated and non-conjugated PUFAs (Supplementary Fig. 2f). By contrast, jacaric- and catalpic acid-induced death was suppressed by fer-1 (Supplementary Fig. 2g). Similar results were found in BT-549 cells (Supplementary Fig. 2h), demonstrating the conserved ability of conjugated 18:3 fatty acids to trigger ferroptosis.

To validate the relative potency of these fatty acids as inducers of cell death, we profiled their effects on viability across 100 human cancer cell lines at 6 doses using the PRISM screening platform[34]. Thunor[35] was used to fit dose-response curves and the area over the curve (Activity Area) was calculated to reflect the potency of cell killing[35]. A histogram reflecting the range of Activity Areas across the cell line panel for each fatty acid is presented as a violin plot in Fig. 3b. Jacaric and catalpic acid were consistently most toxic to the panel, followed by α-calendic acid, αESA and βESA. Whether the differential potency of these conjugated linolenic acids is related to their metabolism by specific enzymes or due to differences in solubility or other physicochemical properties is an important open question.

Punicic acid and arachidonic acid exhibited substantially less anti-proliferative activity and the remaining control fatty acids were largely inactive. These results support a unique death-inducing activity of conjugated linolenic acids.

The tissue of origin of cell lines did not appear to correlate with αESA sensitivity or resistance. However, the sensitivity of individual cell lines to one conjugated linolenate was correlated with sensitivity to the others, but not with control fatty acids (Fig. 3c). Taken together these results show that conjugated linolenic acids exhibit broad toxicity against multiple cancer cell lines of distinct tissue origin. Furthermore, sensitivity or resistance to conjugated linolenic acids is intrinsic to each cell line.

**The role of GPX4 in αESA-induced ferroptosis**. Next, we sought mechanistic insights into ferroptosis induction by αESA. One possibility is that αESA triggers ferroptosis by inhibiting GPX4 directly or indirectly by depleting glutathione. Unlike BSO, treatment with αESA did not significantly affect total cellular glutathione levels despite decreasing cell viability (Fig. 4a). Direct GPX4 inhibitors have been shown to reduce GPX4 protein levels[17]. We confirmed this for ML162 and RSL3 but found that αESA or the negative control erastin had no effect on GPX4 levels (Fig. 4b), suggesting that αESA does not directly inhibit GPX4. As an alternative approach to test if αESA inhibits GPX4, we compared the kinetics of ferroptosis in MDA-MB-231 cells treated with αESA, ML162, and RSL3 alone and in combination. Combining both GPX4 inhibitors at 500 nM triggered fer-1-suppressible cell death with similar kinetics to either GPX4 inhibitor alone, indicating this was a saturating dose of inhibitor (Fig. 4c). In contrast, cell viability was lost significantly more rapidly when cells were co-treated with 500 nM ML162 and 50 μM αESA, a dose of αESA that took longer to induce cell death than either of the GPX4 inhibitors alone. Consistent with ferroptosis, cell death could be rescued by fer-1 (green in Fig. 4c). The ability of αESA to enhance the kinetics of cell death when GPX4 is maximally inhibited shows that αESA and GPX4 inhibitors trigger ferroptosis by distinct mechanisms.

αESA enhanced the toxicity of GPX4 inhibitors in multiple cell lines (Fig. 4d and Supplementary Fig. 3), suggesting that the reduction of hydroperoxides by GPX4 opposes αESA-induced ferroptosis. Consistent with this model, transient knockdown of GPX4 by siRNA also enhanced cell killing by αESA (Fig. 4e). Finally, we generated MDA-MB-231 and BT549-derived cell lines stably overexpressing GPX4 or eGFP as control (Fig. 4f) and found that cell lines overexpressing GPX4 were resistant to

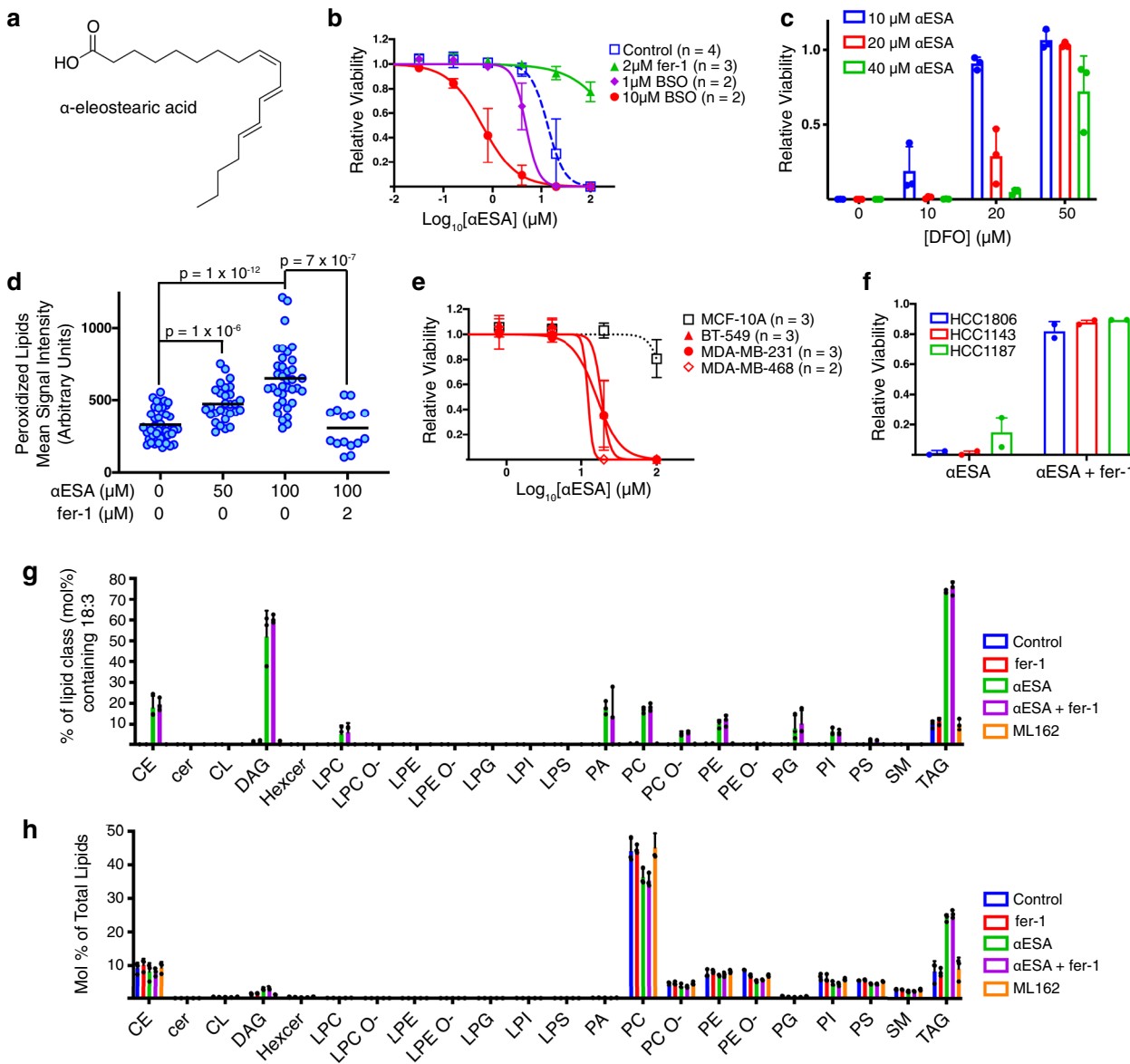

**Fig. 2 The conjugated linolenic fatty acid α-eleostearic acid is incorporated into cellular lipids and induces ferroptosis. a** Structure of α-eleostearic acid (αESA). **b** Cell viability dose-response curves for αESA and the specified additional compound in MDA-MB-231 cells. Cells were treated for 72 h. Error bars in this and subsequent panels represent standard deviation centered on the mean. **c** Relative viability of MDA-MB-231 cells incubated with the indicated doses of αESA and DFO for 72 h ($n = 3$ independent experiments). Values were normalized to account for loss of viability associated with DFO. **d** Quantitation of lipid peroxidation products in individual cells after 4 h of treatment with the specified agent. The line indicates the mean. From left to right, $n = 42, 29, 36,$ and 15. $p$ values from two-sided Student's t tests are shown. **e** Cell viability dose-response curves for αESA for three TNBC cell lines (red lines) and non-cancerous MCF-10A controls (black dashed line) after 72 h of treatment. **f** Relative cell viability for three TNBC cell lines after 72 h of treatment with αESA or the combination of αESA and fer-1 (2 μM) ($n = 2$ independent experiments). The dose αESA was 100 μM for HCC1806 and HCC1143 and 20 μM for HCC1187. **g** Percent of the mole fraction for each of the indicated classes of lipid that contain 18:3 (18 carbon, 3 double bonds) fatty acids consistent with αESA ($n = 3$ biological replicates for each condition). **h** Mole percent of each lipid class as a fraction of total lipids in MDA-MB-231 cells incubated with vehicle, 2 μM fer-1, 50 μM αESA, 50 μM αESA and 2 μM fer-1, or 250 nM ML162 for 3 h ($n = 3$ biological replicates for each condition). CE = cholesterol esters, Cer = ceramide, DAG = diacylglycerol, Hexcer = hexosylceramide, LPC = lysophosphatidylcholine, LPC O- = ether-linked lysophosphatidylcholine, LPE = lysophosphatidylethanolamine, LPE O- = ether-linked lysophosphatidylethanolamine, LPG = lysophosphatidylglycerol, LPI = lysophosphatidylinositol, LPS = lysophosphatidylserine, PA = phosphatidate, PC = phosphatidylcholine, PC O- = ether-linked phosphatidylcholine, PE = phosphatidylethanolamine, PE O- = ether-linked phosphatidylethanolamine, PG = phosphatidylglycerol, PI = phosphatidylinositol, PS = phosphatidylserine, SM = sphingomyelin, TAG = triacylglycerol. Source data are provided as a Source Data file.

ML162-triggered ferroptosis (Fig. 4g) and αESA-induced ferroptosis, though the resistance to ferroptosis triggered by αESA was less robust in BT-549 cells (Fig. 4h).

FSP1 mediates a second pathway for hydroperoxide neutralization that operates in parallel to GPX4. By reducing coenzyme $Q_{10}$, FSP1 maintains an active pool of this radical-trapping lipophilic antioxidant[9,10]. As expected, iFSP1, a small-molecule inhibitor of FSP1[9], enhanced cell killing by ML162 in both BT-549 and MDA-MB-231 cells (Fig. 4i). By contrast, iFSP1 had no effect on αESA toxicity in BT-549 cells, but enhanced cell death in MDA-MB-231 cells (Fig. 4j), highlighting a further distinction between ferroptosis induced by αESA and canonical ferroptosis inducers.

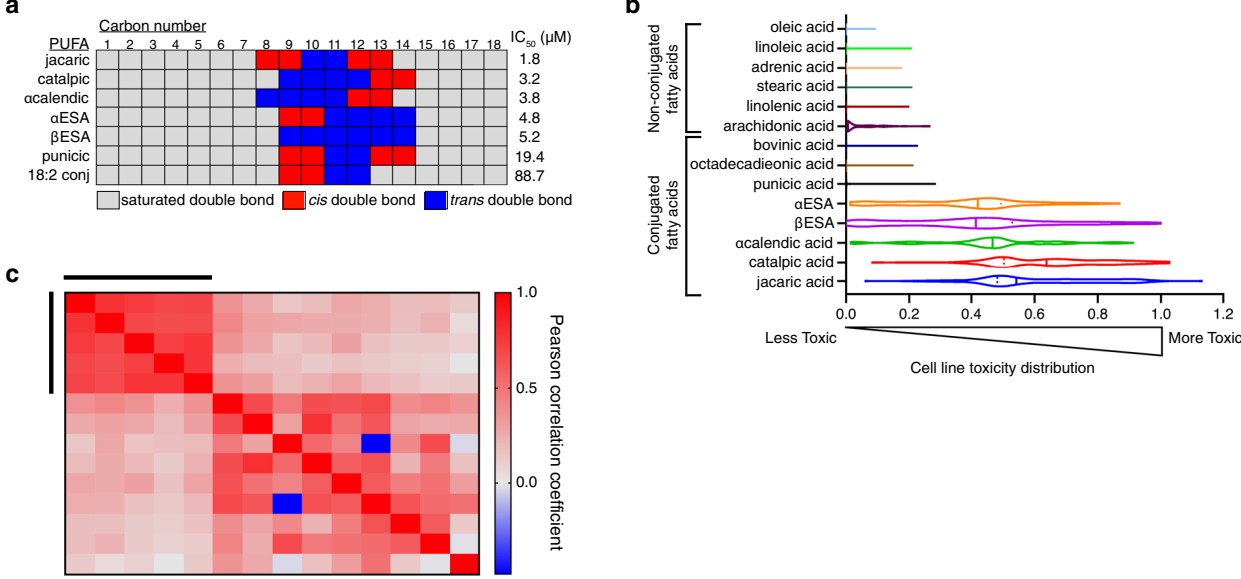

**Fig. 3 Conjugated linolenic fatty acids are generally toxic to diverse cancer cells. a** Schematic representing the location and double bond geometry of 18-carbon conjugated polyunsaturated fatty acids. Cell viability $IC_{50}$ values are shown for each conjugated PUFA from at least two biological replicate measurements in MDA-MB-231 cells after 72 h of treatment. **b** Assessment of fatty acid toxicity across cancer cell lines. A mixture of 100 barcoded human cancer cell lines were treated in triplicate with the indicated fatty acid at doses from 0–35 µM. Viable cells of each cell line in the mixture were quantified at 48 h of treatment. A violin plot illustrates the distribution of cell line-specific toxicities (calculated as the area over the dose dependent-cell viability curve across the treatment dose range). A toxicity value of 1 denotes complete loss of viable cells and 0 corresponds to viability similar to vehicle-treated controls. Dark bars denote median toxicity against all cell lines in the panel and dotted lines indicate quartile boundaries. **c** Heatmap presenting the Pearson correlation coefficients of the activity areas across the cancer cell line panel in **b** for each pair of fatty acids in the test set of fourteen. Red shows positive correlation and blue indicates negative correlation. The bars denote the position of the following conjugated linolenic acids from left to right and top to bottom: jacaric acid, catalpic acid, α-calendic acid, βESA, and αESA. Source data are provided as a Source Data file.

**αESA-induced ferroptosis is ACSL1-dependent**. The first step in esterification of fatty acids into lipids is their conjugation to coenzyme A (CoA), catalyzed by acyl-CoA fatty ligases. There are five long chain-specific isoforms (ACSLs) that are distinguished by fatty acid specificity and subcellular localization[36]. ACSL4 preferentially esterifies certain PUFAs including arachidonic acid and loss of ACSL4 protects cells from ferroptosis induced by direct or indirect inhibition of GPX4 (Fig. 5a)[37–39]. αESA, however, was similarly toxic to *ACSL4*-deficient Pfa1 cells and control parental cells (Fig. 5b), supporting a difference in the mechanism of action of αESA and canonical ferroptosis inducers. αESA-induced cell death was rescued by fer-1 in both cell lines, consistent with ferroptosis (Supplementary Fig. 4a). To identify ACSL isoforms that contribute to αESA-induced ferroptosis, we individually knocked down expression of each of the five isoforms in BT-549 and MDA-MB-231 cells by RNAi (Supplementary Fig. 4b, c). As expected, ML162 toxicity was suppressed by *ACSL4* depletion in both cell lines (Fig. 5c and Supplementary Fig. 4d–f). αESA toxicity, however, was significantly suppressed by knockdown of *ACSL1*, though other ACSL isoforms may also contribute.

We used CRISPR/Cas9 to generate multiple clonal BT-549 cell lines deficient in *ACSL1* (Fig. 5d). ACSL1 loss reduced αESA sensitivity compared to controls (Fig. 5e) and increased sensitivity to ML162 (Fig. 5f). Similar results were obtained using an independent guide RNA (Supplementary Fig. 4g, h). Re-expressing CRISPR-resistant *ACSL1* in *ACSL1*-deficient cells restored αESA sensitivity (Fig. 5g), and increasing *ACSL1* expression in control BT-549 cells further enhanced αESA toxicity (Fig. 5h). These results demonstrate that ACSL1 promotes αESA-triggered cell death. Furthermore, they highlight a mechanistic distinction between αESA and canonical ferroptosis

inducers and expand the spectrum of ACSL isoforms involved in ferroptosis.

**ACSL1 regulates αESA metabolism**. To elucidate the impact of ACSL1 on lipid metabolism, we conducted a lipidomic analysis of two independent *ACSL1*-deficient cell lines as well as an *ACSL1*-overexpressing line and their parental control cells (Fig. 6). Overall, the pattern of lipid enrichment or depletion in response to ACSL1 modulation was generally similar in the absence or presence of αESA (compare Fig. 6 left and middle panels). Among the differentially expressed lipids, we noted that diacylglycerols (DAGs) and triacylglycerols (TAGs) were significantly reduced in *ACSL1*-deficient cells compared to controls or *ACSL1*-overexpressing cells treated with αESA (Supplementary Data 2). These findings are consistent with prior studies linking ACSL1 to the production of these lipids[40,41].

Next, we examined the impact of ACSL1 specifically on lipids bearing 18:3 acyl chains (Fig. 6, right). The mole fraction of 18:3-containing lipids as a class was modestly but significantly decreased in αESA-treated *ACSL1^{-/-}* cells compared to αESA-treated controls ($p = 0.006$, Fisher's combined test). This suggests that the resistance of *ACSL1*-deficient cells to αESA could be related to decreased αESA incorporation. Lipid subgroup-focused analysis revealed 18:3 decreases specifically in phosphatidylglycerols, DAGs, and TAGs (Supplementary Data 3). Conversely, we observed a significant increase in 18:3-bearing cholesterol esters and vinyl ether-linked phosphatidylcholine and phosphatidylethanolamine in αESA-treated *ACSL1*-deficient cells. A potential explanation is that αESA may also be weakly utilized by other ACSL isoforms such as ACSL4, which is known to direct substrates into cholesterol esters[40]. Similar and more robust

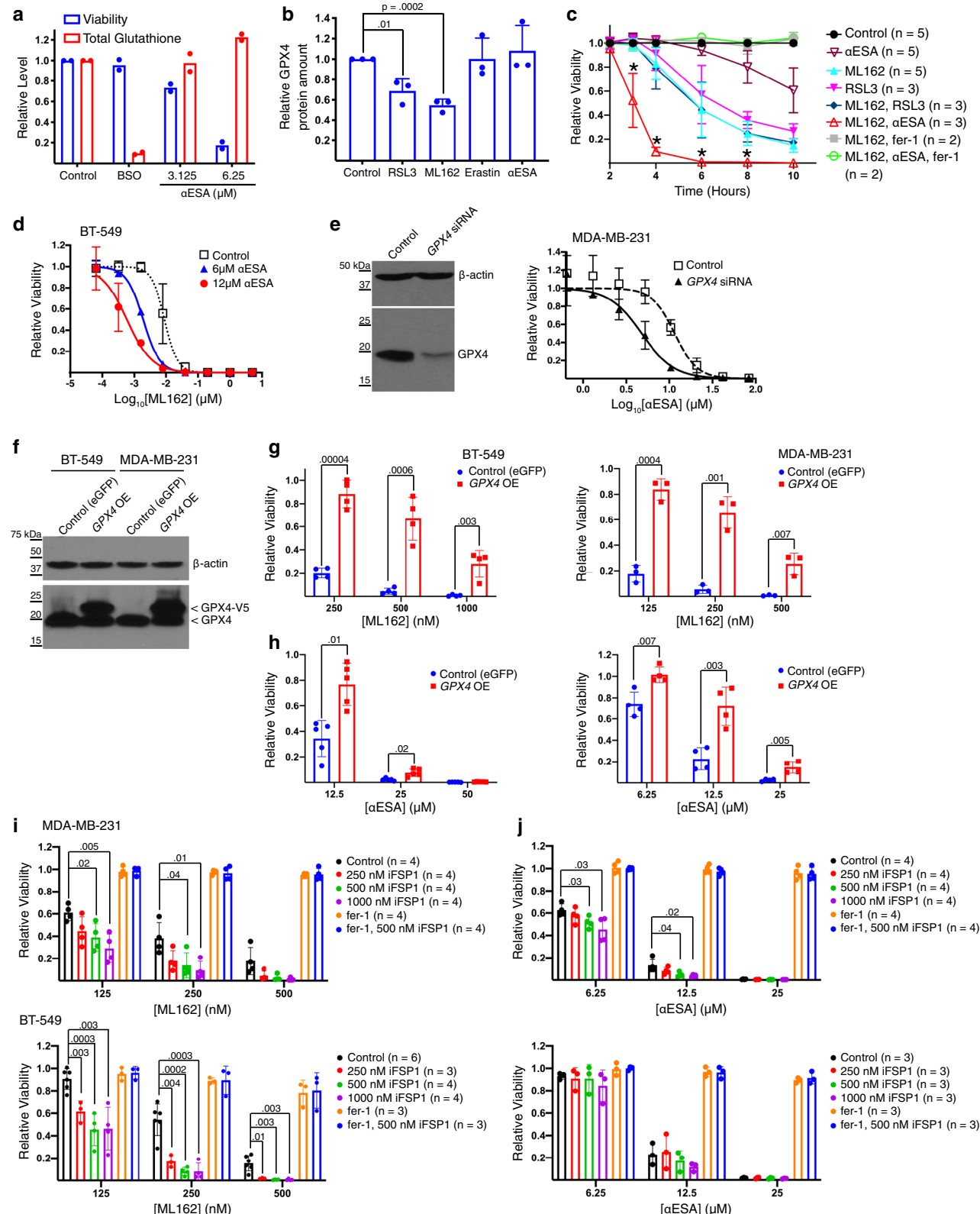

differences were observed comparing *ACSL1*-deficient to *ACSL1*-overexpressing cells. They included significantly lower 18:3-bearing lipids overall ($p = 0.0002$, Fisher's combined test) and particularly in DAGs ($p = 1 \times 10^{-10}$) and TAGs ($p = 0.0002$). In an independent experiment, we confirmed that transient silencing of *ACSL1* expression (Fig. 7a) reduced 18:3 chain incorporation into TAG species of ESA-treated cells (Fig. 7b). Together these

findings demonstrate a role for ACSL1 in the incorporation of αESA and other fatty acids into DAGs/TAGs, which could suggest a role for lipid droplets in αESA-induced ferroptosis.

**αESA triggers ACSL1-dependent increases in lipid hydroperoxides.** We next sought to characterize αESA-induced lipid

**Fig. 4 Mechanistic analysis of αESA-induced ferroptosis. a** Relative viability (blue) and levels of total glutathione (red) in MDA-MB-231 cells after 24 h of treatment with 20 μM BSO, or the specified dose of αESA ($n = 2$ independent experiments). **b** Densitometric quantification of GPX4 protein levels from western blots of lysates from MDA-MB-231 cells that were treated for 4 h with the indicated agent. $n = 3$ independent experiments. Error bars in this and subsequent panels represent standard deviation centered on the mean. $p$ values in this and subsequent panels were calculated using two-sided Student's t tests. **c** Cell viability over time for MDA-MB-231 cells treated as specified. * denote $p$ values comparing viability of cells treated with ML162 and RSL3 to cells treated with αESA and ML162. $p$ values from left to right are 0.003, 0.0001, 0.02, and 0.008. **d** Cell viability dose-response curves for ML162 and the indicated dose of αESA in BT-549 cells after 72 h of treatment ($n = 2$ independent experiments). **e** Western blot showing GPX4 protein level in MDA-MB-231 cells 48 h after transfection with a pool of *GPX4*-targeted siRNA or non-targeting siRNA. β-actin is the loading control for this and the subsequent western blots unless otherwise specified. The right panel shows cell viability dose–response curves for αESA in GPX4-depleted and control MDA-MB-231 cells ($n = 3$ independent experiments). αESA was added to cells 48 h after transfection for 24 h. **f** Western blot of endogenous GPX4 and exogenously expressed V5-tagged GXP4 in BT-549 and MDA-MB-231 cells. Relative viability of these cells after 48 h incubation with the indicated dose of **g**, ML162 ($n = 4$ and 3 for BT-549 and MDA-MB-231, respectively) or **h**, αESA ($n = 5$ and 4 for BT-549 and MDA-MB-231, respectively). $p$-values from Student's t-tests are shown. Relative viability of MDA-MB-231 or BT-549 cells after 48 h of treatment with the indicated dose of either **i**, ML162 or **j**, αESA and vehicle, iFSP1, fer-1 (2 μM), or iFSP1 and fer-1. Source data are provided as a Source Data file.

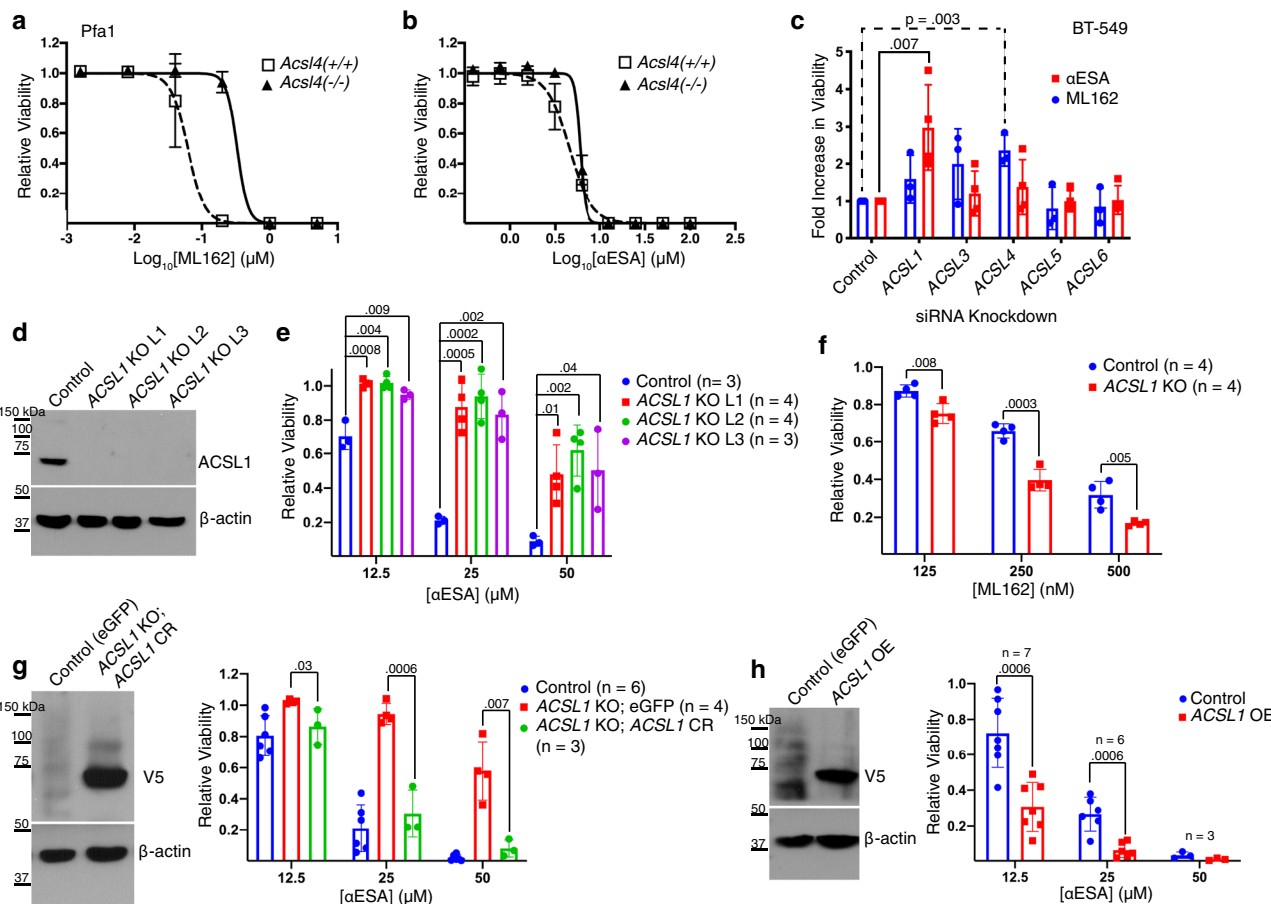

**Fig. 5 ACSL1 mediates ferroptosis triggered by αESA.** Cell viability dose-response curves for control and *ACSL4*-deficient Pfa cells treated with **a**, ML162 or **b**, αESA. Error bars indicate standard deviation from three independent experiments centered on the mean. **c** Bar charts showing fold change in the fraction of viable BT-549 relative to cells transfected with a non-targeting siRNA, 72 h after transfection with the designated *ACSL1*-targeted siRNA followed by 24-h treatment with either ML162 (blue, $n = 3$ independent experiments) or αESA (red, $n = 4$ independent experiments). Error bars in this and subsequent panels show standard deviation centered on the mean. $p$ values < 0.05 from Student's t-test (one-sided) are indicated. **d** Western showing ACSL1 protein levels in control BT-549 cells expressing a non-targeting guide RNA and three single-cell BT-549 clones in which *ACSL1* was disrupted using CRISPR/Cas9 technology (ACSL1 KO L1–3). Similar results were observed using an independent *ACSL1*-targeting guide RNA (Supplementary Fig. 4g). **e** Relative cell viability of control and *ACSL1* KO lines after 48 h of treatment with the specified dose of αESA or **f**, ML162 ($n = 4$ independent experiments). $p$-values are shown above comparator bars in this and subsequent panels (two-sided Student's t-test). **g** V5 western blot (left) showing transgenic re-expression of V5-tagged, CRISPR-resistant *ACSL1* (*ACSL1* CR) and (right) relative viability of control (eGFP), *ACSL1* KO (eGFP), or *ACSL1* KO cells expressing *ACSL1* CR after 48 h of treatment with the indicated dose of αESA. **h** Western blot (left) of V5-tagged ACSL1 in BT-549 cells stably over-expressing the protein compared to a control line expressing eGFP, and (right) the fraction of viable cells remaining for each cell line after 48 h of treatment with the noted dose of αESA. Source data are provided as a Source Data file.

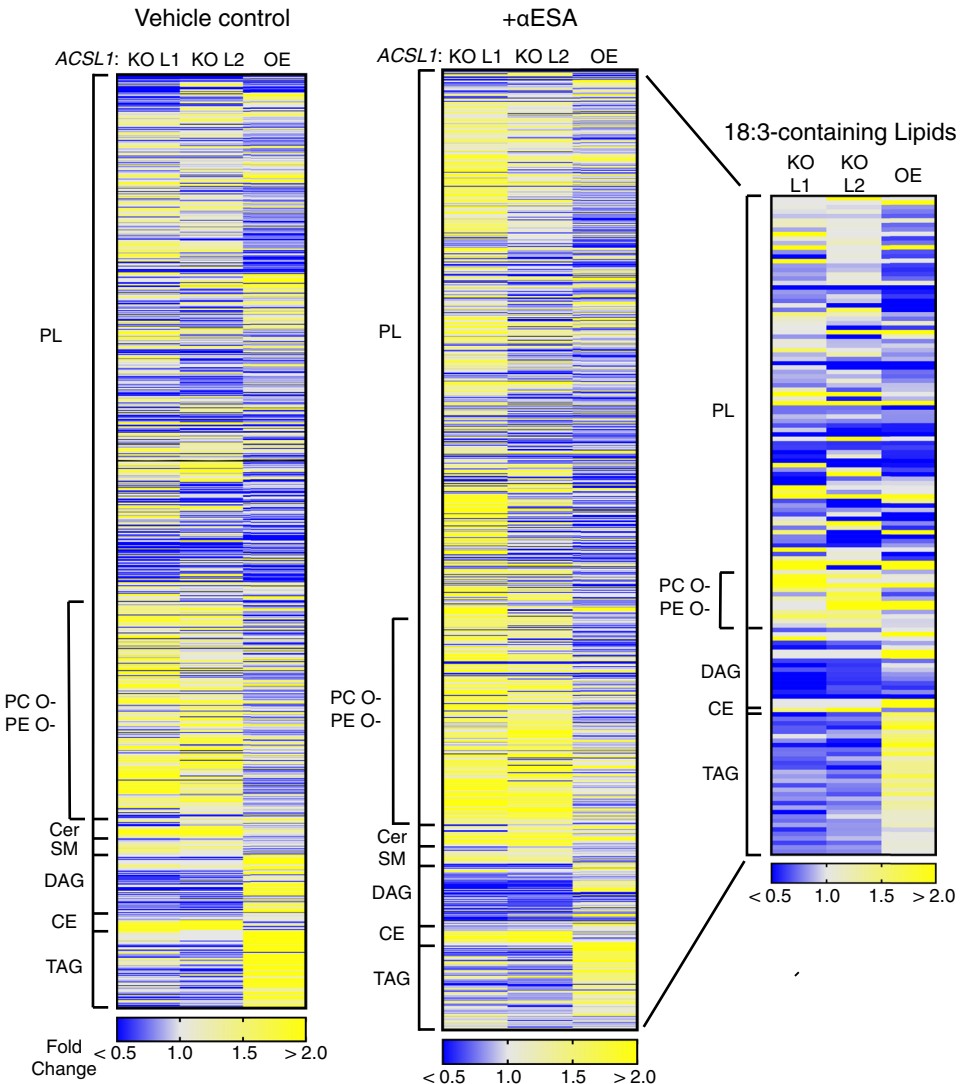

**Fig. 6 ACSL1 drives incorporation of αESA into neutral lipids.** Heatmaps representing relative lipid abundance in *ACSL1*-deficient (*ACSL1* KO L1, L2) or overexpressing (*ACSL1* OE) BT-549 cells as fold-change in mole percent compared to control cells after 16 h treatment with 50 μM αESA (right and middle) or in untreated cells (left). The rightmost heatmap shows only the subset of lipids with 18:3 acyl chains. Yellow indicates an increase in relative abundance and blue shows a decrease in relative abundance. PL = phospholipids, CE = cholesterol esters, Cer = ceramide, DAG = diacylglycerol, PC O- = ether-linked phosphatidylcholine, PE O- = ether-linked phosphatidylethanolamine, SM = sphingomyelin, TAG = triacylglycerol. Source data are provided as a Source Data file.

oxidation. First, we quantified 141 mono-, di-, and tri- oxygenated lipids using liquid chromatography–mass spectrometry across seven membrane lipid classes in BT-549 cells and found the abundance of oxidized membrane lipids was significantly increased in cells treated with αESA ($p = 0.02$, Student's t-test, one-tailed) (Fig. 7c). A wide variety of oxygenated lipids have been previously reported to be increased in cells undergoing ferroptosis, although a key role for phosphatidylethanolamine species bearing arachidonic or adrenic acids has been reported[17]. Within the seven lipid classes examined, we observed that 51 (36%) oxygenated lipid species were significantly changed by αESA and a majority of those were increased (41/51, 80%) (Fig. 7c). These results are consistent with our observation of increased lipid peroxidation products in αESA-treated cells (Fig. 2d). The classes with both the highest number and frequency of increased oxidized species were cardiolipins (17/36 47%) and phosphatidylethanolamines (18/40 45%) and included two di-oxygenated phosphatidylethanolamine species containing arachidonate residues (Fig. 7d, Supplementary Fig. 5).

Of the 41 oxygenated membrane lipids that were significantly increased following incubation with αESA, 10 (24%) were responsive to *ACSL1* silencing and all were decreased (Fig. 7e; Supplementary Fig. 5c, e, g, h). Some, though not all, αESA-induced, di-oxygenated phosphatidylethanolamine lipids were decreased on *ACSL1*-silencing (Fig. 7d). Given the abundant incorporation of αESA in TAGs, we also examined oxidation of this lipid class. Notably, in αESA-treated but not control cells we detected oxygenated 52:4 TAG (18:1/18:3/16:0) species including mono- and di-oxygenated species (Fig. 7f, g), consistent with a prior report[28]. However, few other oxidized TAGs were detected (Supplementary Data 4). The abundance of these species was decreased by ACSL1-silencing, as was the parental, unoxidized 52:4 TAG precursor (Supplementary Fig. 5i), demonstrating that these oxidized TAG species are generated in an αESA and ACSL1-dependent manner.

**Triglycerides contribute to αESA toxicity.** ACSL1 promotes αESA accumulation in DAGs and TAGs (Fig. 6) as well as αESA

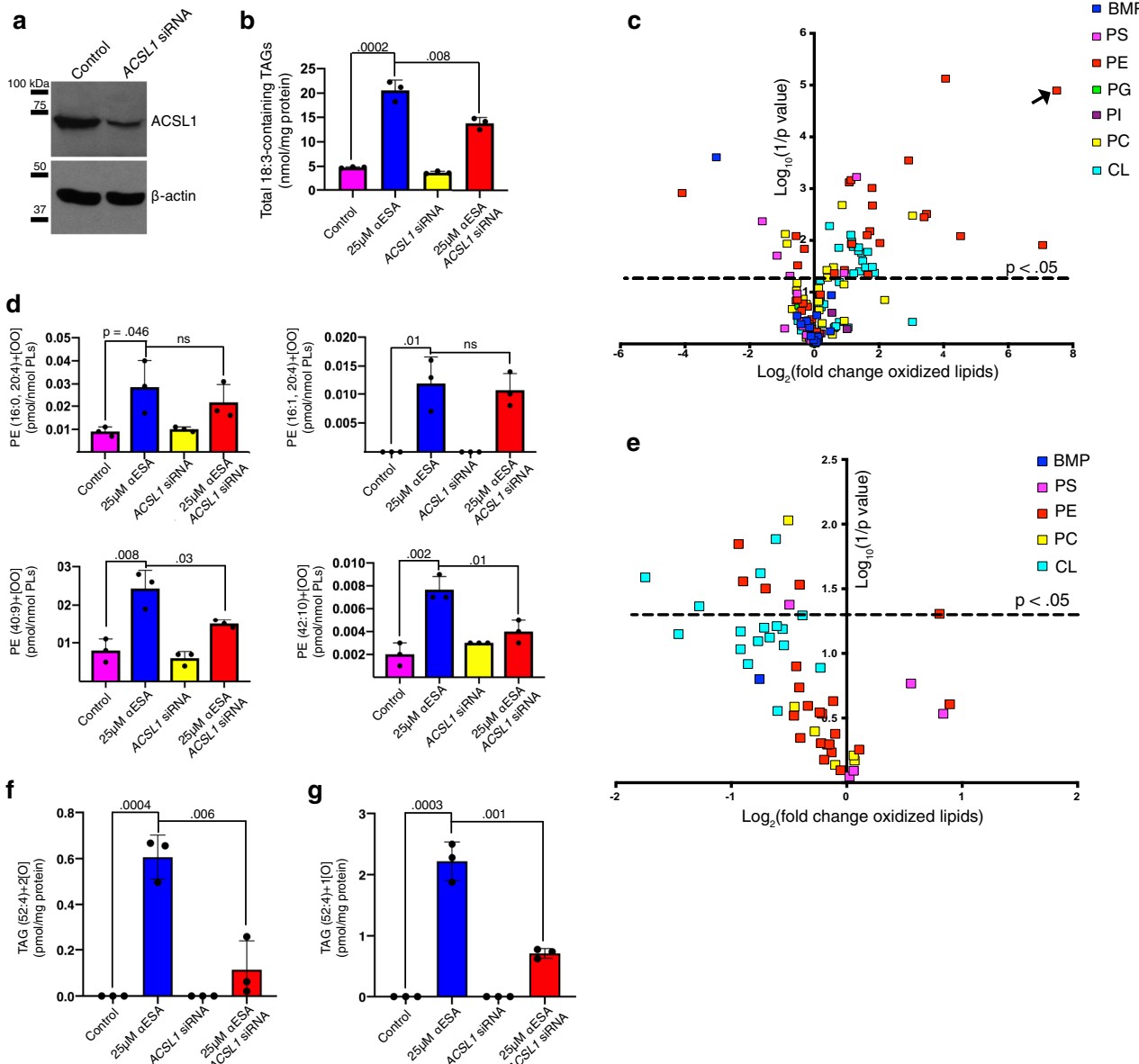

**Fig. 7 Lipid peroxidation by αESA is promoted by ACSL1. a** Western blot showing ACSL1 protein expression in control or *ACSL1*-silenced cells. The efficiency of the siRNA knockdown was independently confirmed using quantitative PCR (qPCR) (Supplementary Fig. 4b). **b** Bar charts showing the overall amount of 18:3-containing TAGs in BT-549 cells under the specified condition ($n = 3$ biological replicates for this and subsequent bar charts). Cells were treated for 8 h with 25 μM αESA or vehicle. Treatments were added 72 h after transfection with a pool of either *ACSL1*-targeting or non-targeting siRNA. In this and subsequent panels, mean values ± standard deviation are presented, and p-values are shown above comparator bars (two-sided Student's t-test). **c** Volcano plot showing log₂(fold change) and significance (log₁₀(1/p)) for oxidized phospholipids in BT-549 cells incubated for 8 h with 25 μM αESA compared to cells incubated with vehicle (methanol). Each lipid species is colored according to phospholipid class. BMP = bis(monoacylglycero)phosphate, PS = phosphatidylserine, PE = phosphatidylethanolamine, PG = phosphatidylglycerol, PI = phosphatidylinositol, PC = phosphatidylcholine, CL = cardiolipin. The arrow indicates a lipid species for which a fold change could not be computed because it was only detected after αESA treatment. **d** Bar charts showing the amounts of di-oxygenated PE species that were significantly increased by αESA treatment compared to vehicle-treated controls. **e** Volcano plot depicts log₂(fold change) versus log₁₀(1/p-value) for oxidized phospholipids in BT-549 cells transfected with a pool of *ACSL1* siRNA and incubated with 25 μM αESA for 8 h compared to αESA-treated cells transfected with non-targeting siRNA. The set is limited to the 51 oxidized phospholipid species that were significantly changed by αESA treatment as shown in **c**. Bar charts showing the amount of a (**f**), di-oxygenated and (**g**), mono-oxygenated TAG species that were significantly increased by αESA treatment in an ACSL1-dependent manner. Source data are provided as a Source Data file.

toxicity (Fig. 5), suggesting a potential mechanistic link. We, therefore, examined the role of diacylglycerol acyl transferase (DGAT), which acylates DAGs to produce TAGs, in αESA toxicity. Transient silencing of *DGAT1* but not *DGAT2* restored viability of αESA-treated MDA-MB-231 cells (Fig. 8a, Supplementary Fig. 6a, b). Likewise, treatment of cells with the

DGAT1-specific inhibitor PF-04620110 but not the DGAT2-specific inhibitor PF-06424439 suppressed αESA toxicity (Fig. 8b). Consistent with redundancy, the combination of both inhibitors produced an additive effect in MDA-MB-231 cells (Fig. 8b) and this combination also suppressed αESA-induced cell death in BT-549 cells (Fig. 8c). ML162 toxicity was not affected by

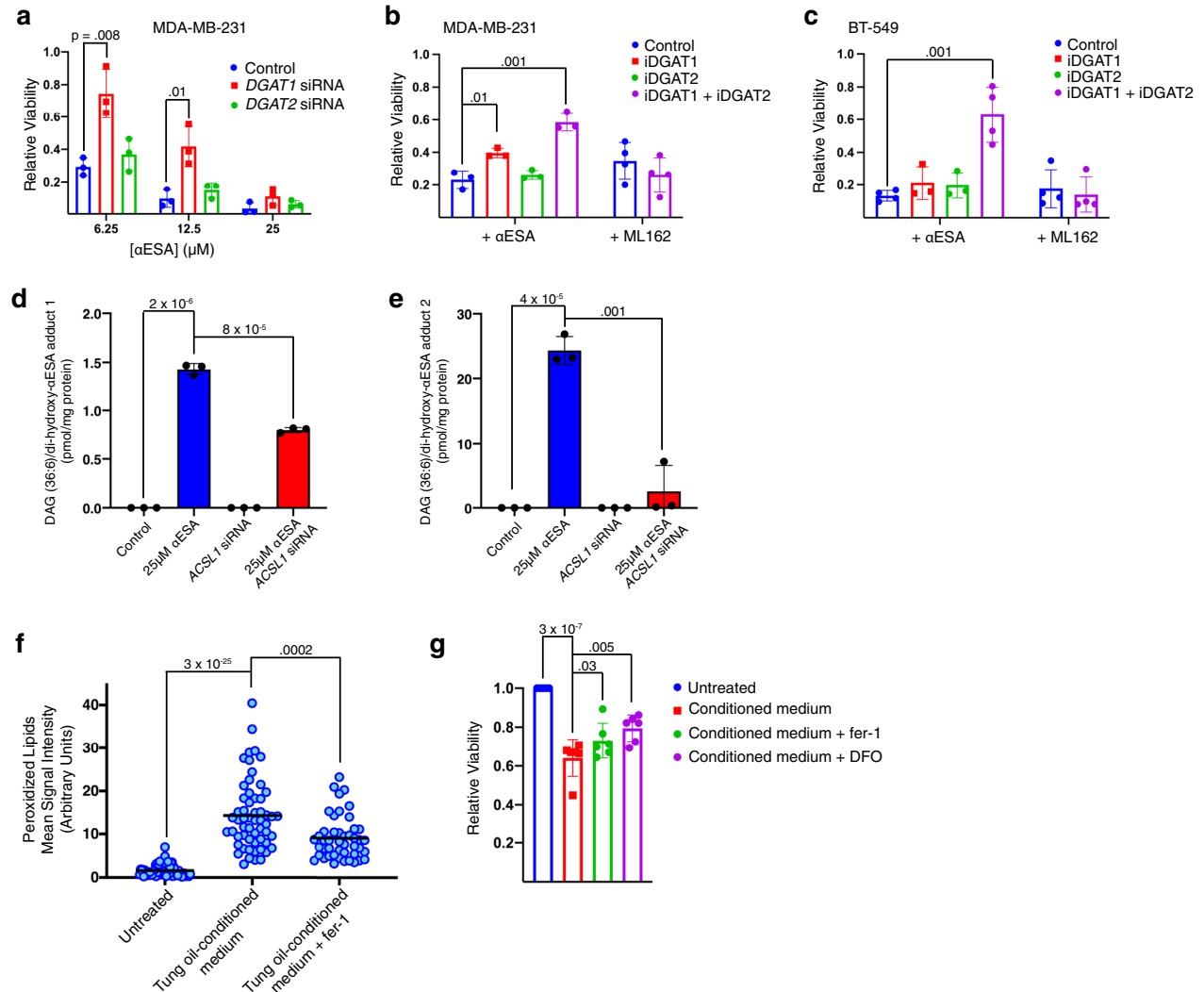

**Fig. 8 Incorporation of αESA into triacylglycerols promotes ferroptosis. a** Relative viability of MDA-MB-231 cells 72 h after transfection with either *DGAT1*-targeting, *DGAT2*-targeting, or non-targeting siRNA followed by 24-hour treatment with indicated dose of αESA ($n = 3$ independent experiments). Error bars in this and subsequent panels show standard deviation centered on the mean, and p-values above comparator bars in this and subsequent panels are from two-sided Student's t-tests unless noted. Bar charts showing the fraction of viable (**b**), MDA-MB-231 or (**c**), BT-549 cells incubated for 48 h with either 6.25 μM αESA ($n = 3$ and 4 for MDA-MB-231 and BT-549, respectively) or 62.5 nM ML162 ($n = 4$) and either vehicle, a DGAT1 inhibitor (4 μM PF-04620110), a DGAT2 inhibitor (4 μM PF-06424439), or both inhibitors. **d, e** Bar charts showing the amount of two DAG (18:3/18:3)/dihydroxy αESA adducts that were detected following treatment with αESA and were significantly decreased by *ACSL1* depletion ($n = 3$ biological replicates). The m/z of adduct 1 and adduct 2 are 940.7278 and 942.7439, respectively. **f** Quantitation of lipid peroxidation products in individual BT-549 cells after being cultured for 2 h with normal medium, tung oil-conditioned medium, or tung oil-conditioned medium containing 2 μM fer-1. The line indicates the mean. From left to right, $n = 71$, 57, and 46. **g** Bar charts showing relative cell viability for BT-549 cells cultured in normal growth medium or tung oil-conditioned medium with or without 2 μM fer-1 or 50 μM DFO ($n = 6$ independent experiments). The values above the comparator bars represent the p-values from one-sided Student's t-tests. Source data are provided as a Source Data file.

this inhibitor combination (Fig. 8b, c) indicating a role for TAGs specifically in αESA-induced ferroptosis. Knockdown of adipose triglyceride lipase (ATGL; Supplementary Fig. 6c), which liberates fatty acids from TAGs, suppressed the toxicity of ML162 but not αESA in two cell lines (Supplementary Fig. 6d–g). Atglistatin, a pharmacological inhibitor of ATGL, in contrast, suppressed both αESA and ML162 toxicity (Supplementary Fig. 6h–j), suggesting the possibility that atglistatin might act as a direct radical trapping agent as has been observed for other small molecule inhibitors[42].

Fatty acids and TAGs with conjugated double bonds undergo complex and poorly understood oxidative reactions distinct from that of unsaturated chains with non-conjugated double bonds[43–47]. Products of conjugated acyl chain oxidation include

aldehydes, peroxides, and high molecular weight covalent polymers[45–49]. Our mass spectrometry analysis of low molecular weight lipids from αESA-treated cells revealed the formation of two distinct adducts of di-oxygenated αESA free fatty acid with DAG containing two αESA-derived chains (Fig. 8d, e). These adducts were only detected in αESA-treated cells and were significantly reduced in *ACSL1*-knockdown cells, demonstrating an ACSL1-dependent formation of oxidative adducts of αESA-derived acyl chains in neutral lipids. Higher-order polymers or adducts of αESA-containing TAGs were beyond the size range analyzed and thus our analysis likely underestimates the scope of TAG oxidation.

A remaining question is whether oxidative decomposition of αESA-derived acyl chains in neutral lipids could lead to the

oxidation of membrane phospholipids. The seed oil of the tung tree is composed of TAGs in which ~80% of the acyl chains are esterified αESA. We, therefore, considered tung oil as a model for the αESA-rich TAGs that accumulate in αESA-treated cells (Fig. 2g). We generated an emulsion of tung oil in cell culture medium and, following sedimentation and removal of the oil layer, incubated the conditioned medium with BT-549 cells. Two hours in conditioned medium led to an increase in cellular lipid peroxidation products, which were suppressed by co-treatment with fer-1 (Fig. 8f). Incubating cells with conditioned medium for 24 h resulted in significantly decreased cell viability, which was partially rescued by fer-1 or iron chelation by deferoxamine (Fig. 8g). These findings demonstrate that αESA-rich triglycerides can produce a diffusible mediator of ferroptosis that could potentially propagate lipid peroxidation from neutral lipids to membrane lipids.

**Conjugated PUFAs exhibit anti-cancer activity and promote expression of ferroptotic markers in vivo.** Finally, we assessed the anti-cancer activity of tung oil in an aggressive TNBC orthotopic xenograft model. Mice with established tumors were treated by oral gavage five times per week with 100 µl tung oil. Control mice received high-oleic (18:1) safflower oil. Body weights of tung oil-treated mice did not differ significantly from controls (Supplementary Fig. 7a). However, tung oil significantly suppressed tumor growth (Fig. 9a) and endpoint tumor weight (Fig. 9b) and also substantially reduced lung metastatic invasion compared to controls (Fig. 9c, d). The addition of BSO further enhanced tumor growth suppression by αESA (Supplementary Fig. 7b), consistent with BSO enhancing αESA toxicity in vitro (Fig. 2b).

To obtain evidence that the reduction in tumor growth is associated with tumor exposure to αESA, we conducted a lipidomic analysis of αESA-treated and control tumors. The two most differentially expressed lipids were DAG (34:3) and TAG (52:7), which were increased in tung oil-treated tumors by 30% and 60%, respectively (Fig. 9e). The masses of these lipids are consistent with at least one linolenate moiety, although their precise acyl chain composition was not determined. Notably, the expression of these two lipids was also increased in αESA-treated cultured cells (Fig. 9e). Thus, increased DAG (34:3) and TAG (52:7) could reflect tumor exposure to circulating αESA released from tung oil and taken up by tumor cells.

Next, we assessed whether tumors in αESA-treated mice expressed markers of ferroptosis. We conducted RNA sequencing of three tumors each from treated and control mice and compared the differentially expressed genes with a separate in vitro experiment using cultured MDA-MB-231 cells treated with either αESA, the GPX4 inhibitor ML162, or vehicle controls. Using a false discovery rate of 5% and a filter of >2-fold change in RNA expression, 124 genes were altered in tung oil-treated tumors compared to safflower oil-treated controls (Fig. 9f). Significantly, 89 genes (89/122, 73%) altered in tung oil-treated tumors were also altered in αESA-treated cells in culture. Fisher's exact test showed significant overlap between these gene sets ($p = 3.2 \times 10^{-246}$) and all 89 overlapping genes were altered in the same direction. This is consistent with our hypothesis that the inhibition of tumor growth in vivo is related to the cell death caused by αESA in vitro. Importantly, there was also significant overlap in genes altered by tung oil in vivo, αESA in vitro, and by in vitro treatment with ML162 (Fig. 7f). Fisher's exact test between any two gene sets demonstrated $p$-values of $<7.2 \times 10^{-61}$. Furthermore, all of the shared genes were altered in the same direction. These results demonstrate concordant gene signatures between tung oil treatment and αESA treatment, consistent with

the hypothesis that tumors in tung oil-treated mice are exposed to αESA. Furthermore, the overlap in the signatures of αESA- and ML162-treated cells provide strong evidence that both treatments trigger a similar ferroptotic response.

All three treatments altered a common set of 23 genes (Fig. 9f) including genes previously identified as increased during ferroptosis (*RHOB, SLC2A3, DDIT4,* and *HMOX1*)[50,51]. In addition, the previously reported ferroptosis marker *CHAC1*[50] was commonly upregulated in both tung-oil treated tumors and αESA-treated cells. 66 genes were altered by both αESA and tung oil but not ML162, suggesting a potential source of biomarkers for αESA-induced ferroptosis.

**Discussion**

While anti-cancer activity has been previously attributed to various conjugated linolenic acids including αESA[52,53], here we reveal the mode of action as ferroptosis and highlight mechanistic differences compared to existing ferroptosis inducers. Our findings suggest that while GPX4 inhibitors promote ferroptosis by inhibiting the glutathione-dependent reduction of hydroperoxides, αESA promotes ferroptosis by enhancing hydroperoxide production. GPX4 activity reduces αESA-induced hydroperoxide accumulation but ultimately GPX4 is overwhelmed.

We found that ACSL1 mediates αESA-induced ferroptosis as well as αESA incorporation into specific lipid species including DAGs and TAGs (Figs. 5, 6 and 7b). Further supporting a role for TAGs in ferroptosis, depletion or pharmacological inhibition of DGAT activity suppressed αESA toxicity (Fig. 8a–c). We found that ACSL1, which is constitutively localized to lipid droplets[54], concentrates αESA in neutral lipids compared to other lipid pools. By three hours of treatment with αESA, TAG levels increase 2-fold (Fig. 2h) with >70% containing at least one 18:3 acyl chain while no membrane phospholipid class has more than 20% (Fig. 2g). We hypothesize that neutral lipids bearing αESA-derived acyl chains undergo spontaneous or enzyme-mediated oxidation after achieving a critical concentration in lipid droplets. This concentration dependence could reflect the intermolecular nature of the rate-limiting propagation step of lipid autoxidation. The peroxy radicals that mediate the chain-reaction propagation phase of autoxidation either abstract a hydrogen atom from adjacent hydroperoxides or form intermolecular covalent bonds by adding to carbon-carbon double bonds. The rate of abstraction of a hydrogen atom adjacent to the triene system of conjugated linolenic acid is 2.5-fold faster than the corresponding rate for abstraction of the bis-allylic hydrogen in non-conjugated linolenic acid[43]. Furthermore, peroxyl radicals also add to the triene system of conjugated linolenic acid at an even faster rate while peroxyl radicals do not appreciably add to double bonds in non-conjugated linolenic acid. This results in an overall rate of free radical oxidation propagation that is over 8-fold faster for conjugated linolenic acid[43], providing a chemical basis for the distinct ferroptosis-inducing activity of this lipid class above a threshold concentration.

While we only detected two oxidized TAG species associated with αESA treatment (Fig. 7f, g), we speculate that neutral lipids containing esterified αESA undergo oxidative polymerization to form higher molecular weight polymers, as has been previously shown ex vivo[45–48]. Because we focused our analysis on lipids <1,200 molecular weight, very large species, and any oxygenated derivatives were not examined by our lipidomic analysis. However, our demonstration of adducts of free αESA with DAGs containing esterified αESA (Fig. 8d, e), which fall under this size cutoff, supports the possibility that larger covalent aggregates may also form.

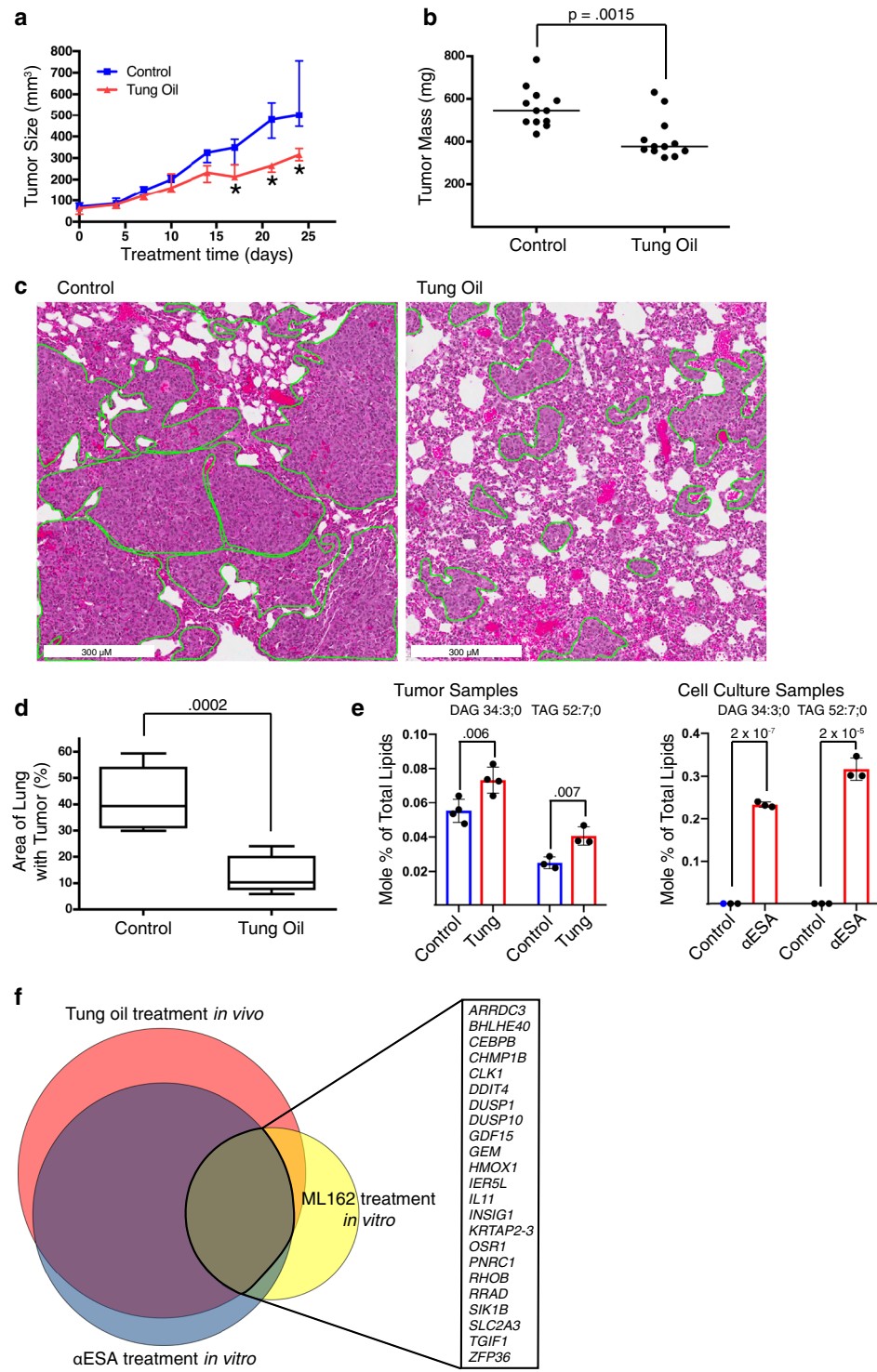

A role for lipid droplets in potentiating lipid peroxidation is contrary to a proposed protective role for this organelle against oxidative stress[55,56]. This may be explained, however, by the rapid rate of free radical oxidation propagation by conjugated linolenic acids. Dietary conjugated linoleic acids have been linked to increased lipid peroxidation in humans[57,58] and animal models[59], and αESA exhibits even more rapid spontaneous oxidation than these species[53,60]. While lipid droplets may protect poly-unsaturated lipids from oxidation by sequestering them from ROS species generated elsewhere, the accumulation of esterified conjugated linolenic acids, which are not naturally synthesized by

mammals, in lipid droplets may promote their oxidation and polymerization there. Propagation of lipid peroxidation from lipid droplets to membrane phospholipids may be mediated by diffusion of soluble mediators (Fig. 8f, g). The formation of large covalent lipid polymers in lipid droplets may also trigger an autophagic response[61] which could drive ferroptosis[62].

A model in which accumulation of conjugated linolenic acyl chains in lipid droplets initiates ferroptosis may also explain the differences in ACSL isoform roles between ferroptosis triggered by αESA and by GPX4 inhibitors. GPX4 inhibitors suppress the antioxidant response, allowing natural lipid ROS species to

**Fig. 9 Tung oil suppresses TNBC xenograft growth and metastasis with markers of ferroptosis. a** Median tumor volumes over time for orthotopic MDA-MB-231 xenografts in NSG mice treated orally with either safflower oil (control) or tung oil ($n = 6$ mice per group with bilateral tumors). * denote p-values from two-sided Student's $t$-test, which are 0.003, 0.00001, and 0.0003 from left to right. Error bars show interquartile range. **b** Final tumor masses of individual tumors following 24 days of treatment. Lines represent median values. The $p$-value above the comparator bar is from a two-sided Student's $t$-test. **c** Representative hematoxylin and eosin-stained lung sections. Regions containing metastatic TNBC cells are outlined in green. **d** Quantification of the percentage of lung area infiltrated by metastatic cells. The line represents the median value, the box shows the interquartile range (25th to 75th percentile), and the whiskers show minimum and maximum values. This analysis was performed once. The p-value above the comparator bar is from a two-sided Student's $t$-test. **e** Mole percent of total lipids for DAG 34:3 and TAG 52:7 in tumors treated with tung oil (left, $n = 4$ and 3 biological replicates for DAG 34:3 and TAG 52:7, respectively) or in MDA-MB-231 cells grown in culture and incubated with 50 μM αESA for 3 h (right, $n = 3$ biological replicates). Error bars represent standard deviation centered on the mean. The p-values were calculated using a one-sided Student's $t$-test, not adjusted for multiple hypothesis testing. **f** Proportional Venn diagram showing the overlap in genes with transcript levels that were significantly altered by >2-fold (5% false discovery rate) after a 5 h incubation with 100 μM αESA or 125 nM ML-162 or in orthotopic MDA-MB-231 xenograft tumors treated as described above for **a**. Genes commonly altered in all three data sets are listed. Source data are provided as a Source Data file.

accumulate. In many contexts these are likely enzymatically generated peroxidation products of arachidonyl and adrenyl acyl chains, whose incorporation into membrane phospholipids is mediated by ACSL4[38]. By contrast, the ACSL1-dependence of αESA toxicity appears to be mediated at least in part by ACSL1-driven incorporation of αESA into TAGs. Overall, these findings suggest that conjugated linolenic acids trigger ferroptosis by a mechanism that is mediated by ACSL1 and support emerging evidence implicating lipid composition as a critical determinant of ferroptotic sensitivity[63].

The discrepant ability of GPX4 and FSP1 to suppress αESA-triggered ferroptosis compared to ferroptosis induced by ML162 in BT549 cells (Fig. 4g–j) could also be explained by distinct origins of lipid peroxidation. We speculate that GPX4 and FSP1 may have evolved and be localized specifically to target natural sources of lipid ROS. Indeed, FSP1 is N-myristoylated and is localized to intracellular membranes and the plasma membrane in a cell type-specific manner[9,10]. Plasma membrane-restricted FSP1 restored RSL3-resistance to FSP1-deficient cells while FSP1 targeted to other intracellular locations did not[10], suggesting that the plasma membrane is an important site of lipid radical accumulation on GPX4 inhibition. αESA, by contrast, is metabolized like other nutrient fatty acids and stored in lipid droplets where, because of its oxidizable conjugated triene system, it may act as a source of lipid ROS that is poorly targeted by endogenous protective mechanisms. This concept has clear applicability to targeting cancer cells that robustly scavenge extracellular fatty acids[22].

Studies have shown that cancer cells, including renal cell carcinoma, melanoma, and glioblastoma are vulnerable to ferroptosis[38,51,64–68]. Furthermore, therapy-resistant mesenchymal cancer cells exhibit a greater reliance on GPX4 activity for survival[64,69]. These findings have highlighted the therapeutic potential of ferroptosis, but it remains unclear whether induction of ferroptosis can be used to kill cancer cells in vivo[1]. A major challenge to testing this hypothesis has been the absence of ferroptosis-inducing agents, such as GPX4 inhibitors, suitable for use in vivo[70]. Here, we found that treatment of mice with tung oil, a rich source of αESA, suppressed tumor growth and metastasis, and resulted in gene expression changes in tumors consistent with ferroptosis. The data suggest that GPX4 inhibitors, should these become clinically viable, may enhance the anti-tumor activity of αESA. Thus, these findings expand our understanding of the pathways that regulate ferroptosis and provide an approach for exploiting ferroptosis therapeutically.

## Methods

**Cell lines**. MDA-MB-468 cells were obtained by the Cell Culture Facility at Fox Chase Cancer Center from ATCC (HTB-132) as part of the NCI-60 panel. MDA-MB-231 (HTB-26), BT-20 (HTB-19), BT-549 (HTB-122), HCC38 (CRL-2314),

HCC1806 (CRL-2335), HCC1187 (CRL-2322), HCC1143 (CRL-2321), HCC70 (CRL-2315), Hs-578T (HTB-126), cell lines and the MCF-10A cell line (CRL-10317) were obtained directly from the American Type Culture Collection (ATCC, Manassas, VA 20110, USA). Note: BT-20 has been identified as a frequently misidentified cell line by the ICLAC Register of Misidentified Cell Lines but ATCC maintains the authentic stock (HTB-19). The suppliers routinely authenticate the cell lines by short tandem repeat profiling though only MDA-MB-231 cell lines were authenticated by our laboratory (April 2018). MCF-12A cells were a kind gift of Dr. Jose Russo. All cell lines were amplified and frozen within 2 months of receipt. TNBC cell lines were cultured in RPMI-1640, 10% heat inactivated fetal bovine serum (FBS), 2 mM supplemental glutamine, and 100 μg/mL penicillin/streptomycin (P/S) with the exception of BT-20, which was cultured in MEM with the same supplements. MCF-10A and MCF-12A cells were cultured in high calcium medium with 5% horse serum as described[71]. Lenti-X 293 T cells were cultured in DMEM plus 10% FBS, 2 mM glutamine, and P/S. The FBS content of the medium was increased to 30% during lentiviral production. Mouse embryonic fibroblast (Pfa1) *Acsl4*-deficient and control cells were cultured as described previously[38]. Cells were cultured in a humidified incubator at 37 °C with 5% $CO_2$. Cell lines are periodically tested for *Mycoplasma* contamination using DAPI (4′,6-diamidino-2-phenylindole) to stain DNA.

**PRISM 100 cell line viability profiling**. Cell viability across a panel of 100 human cancer cell lines at 6 doses for each of 14 fatty acids was determined using the multiplexed PRISM screening platform[34]. Quantitative cell line bar code data from cells treated for 48 h with the indicated fatty acid (in triplicate at 5–6 doses ranging from 0 to 35 μM) were scaled from 0 (no viable cells) to 1 (full viability) based on median values from DMSO-treated control wells and wells treated with saturating doses of the death-inducing control compound bortezomib. Rescaled dose-dependent cell viability data were fitted to a three-parameter log-logistic model and Activity Area (reflecting cell line-specific toxicity) was calculated using Thunor[35] for each combination of fatty acid and cell line. A histogram reflecting the range of Activity Areas calculated across the cell line panel for each fatty acid is presented as a violin plot in Fig. 3b. Pearson correlation coefficients were calculated for the Activity Areas across the cell line panel for each pair of fatty acids. The heatmap shown in Fig. 3c was generated using GraphPad Prism (Version 8).

**Statistical Analysis**. Metabolite profiling data shown in Fig. 1h and i were $\log_{10}$-transformed before further analysis to achieve an approximate normal distribution. Missing values were not imputed for univariate analysis. Mixed model analysis of variance (ANOVA) using R with package nlme was applied to identify differentially expressed intracellular metabolites in non-ferroptotic (HCC38, HCC1806, HCC1143, HCC70) compared to ferroptotic (MDA-MB-468, MDA-MB-231, BT-549, BT-20, Hs-578T) TNBC cell lines. Our previously reported data set of 155 metabolites including 5–6 replicates per cell line[23] was taken considering "cell line" as a random factor and "ferroptosis response" as categoric factor. ANOVA models were read out concerning t-statistics results comprising estimates, t-values, and p-values. Significance level was set to an α-error of 5%. The multiple test problem was addressed by calculating the false discovery rate (FDR) using the Benjamini & Hochberg method. For the lipidomic data presented in Fig. 6 and Supplementary Data 2 and 3, combined p-values were calculated using Fisher's combined probability test, and the significance threshold was corrected using the Benjamini & Hochberg method and 5% FRD. In general, other statistical comparisons were performed using Student's t-test unless noted and the threshold for significance was $p < 0.05$. Data are reported as mean and standard deviation of at least 2–3 independent experiments unless otherwise stated. All data points reflect measurements of distinct samples.

**Lipidomic analysis**. For Fig. 2g, h, triplicate samples of MDA-MB-231 cells at ~60% confluence were incubated with vehicle (methanol), 2 μM fer-1, 50 μM αESA, 50 μM αESA and 2 μM fer-1, or 250 nM ML162 for 3 h and then were typsinized,

washed twice with phosphate buffered saline (PBS) without magnesium or calcium, and resuspended at a concentration of 1500 cells/μL, and frozen in liquid nitrogen. For Fig. 6, three replicates of two independent *ACSL1*-deficient BT-549 cell lines, a stably over-expressing *ACSL1* line, or a control line expressing a non-targeting guide were incubated with either vehicle (methanol) or 50 μM αESA (Cayman Chemical) for 16 h and then prepared as above. Cells of a different passage number were used for each replicate. Plates were approximately 70–80% confluent at the time of harvest. For tumor samples (Fig. 9e), four biological replicate tumor samples per condition were homogenized in PBS without magnesium or calcium using a dounce homogenizer to yield a sample containing 5 mg of tumor (wet weight)/mL and were then frozen in liquid nitrogen.

Mass spectrometry (MS)-based lipid analysis in Figs. 2g, h, 6, and 9e were performed by Lipotype GmbH (Dresden, Germany) as described[72]. Lipids were extracted using a two-step chloroform/methanol procedure[73]. Samples were spiked with internal lipid standard mixture containing: cardiolipin 16:1/15:0/15:0/15:0 (CL), ceramide 18:1;2/17:0 (Cer), diacylglycerol 17:0/17:0 (DAG), hexosylceramide 18:1;2/12:0 (HexCer), lyso-phosphatidate 17:0 (LPA), lyso-phosphatidylcholine 12:0 (LPC), lyso-phosphatidylethanolamine 17:1 (LPE), lyso-phosphatidylglycerol 17:1 (LPG), lyso-phosphatidylinositol 17:1 (LPI), lyso-phosphatidylserine 17:1 (LPS), phosphatidate 17:0/17:0 (PA), phosphatidylcholine 17:0/17:0 (PC), phosphatidylethanolamine 17:0/17:0 (PE), phosphatidylglycerol 17:0/17:0 (PG), phosphatidylinositol 16:0/16:0 (PI), phosphatidylserine 17:0/17:0 (PS), cholesterol ester 20:0 (CE), sphingomyelin 18:1;2/12:0;0 (SM), triacylglycerol 17:0/17:0/17:0 (TAG). After extraction, the organic phase was transferred to an infusion plate and dried in a speed vacuum concentrator. The dried extract was re-suspended in 7.5 mM ammonium acetate in chloroform/methanol/propanol (1:2:4, V:V:V) and the second step dry extract was re-suspended in a 33% ethanol solution of methylamine in chloroform/methanol (0.003:5:1; V:V:V). All liquid handling steps were performed using the Hamilton Robotics STARlet robotic platform. Samples were analyzed by direct infusion on a QExactive mass spectrometer (ThermoFisher Scientific) equipped with a TriVersa NanoMate ion source (Advion Biosciences). Samples were analyzed in both positive and negative ion modes with a resolution of $Rm/z = 200 = 280000$ for MS and $Rm/z = 200 = 17500$ for MS-MS experiments, in a single acquisition. MS–MS was triggered by an inclusion list encompassing corresponding MS mass ranges scanned in 1 Da increments[74]. Both MS and MS-MS data were combined to monitor CE, DAG, and TAG ions as ammonium adducts; PC, PC O-, as acetate adducts; and CL, PA, PE, PE O-, PG, PI and PS as deprotonated anions. MS only was used to monitor LPA, LPE, LPE O- and LPI and LPS as deprotonated anions; Cer, HexCer, SM, LPC, and LPC O- as acetate adducts. Data were analyzed using lipid identification software based on LipidXplorer[75,76]. Data post-processing and normalization were performed using an in-house developed data management system (Lipotype). Only lipid identifications with a signal-to-noise ratio >5, and a signal intensity 5-fold higher than in corresponding blank samples were considered for further data analysis. To approximate the mole percent of triacylglycerol species containing αESA-derived acyl chains, triacylglycerols with at least three desaturations that were significantly increased above the detection limit, or increased greater than 2-fold compared to control cells after αESA treatment, were considered to contain esterified αESA. The heatmap shown in Fig. 6 was generated using GraphPad Prism (Version 8). Lipids below the detection limit were considered to have a mole percent of zero.

### Quantification of oxidized cellular phospholipids and triacylglycerols (TAGs).

BT-549 cells were transfected with *ACSL1* or non-targeting siRNA 72 h prior to incubation with 25 μM αESA or vehicle (methanol) for 8 h. After 8 h, cells (~4–6 × 10[6] per replicate) were trypsinized, collected, washed once with phosphate-buffered saline, and frozen in liquid nitrogen. Triplicate samples were collected for each condition. MS analysis of PLs was performed on an Orbitrap™ Fusion™ Lumos™ mass spectrometer (ThermoFisher Scientific). PLs were separated on a normal phase column (Luna 3 μm Silica (2) 100 Å, 150 × 2.0 mm, (Phenomenex)) at a flow rate of 0.2 mL/min on a Dionex Ultimate 3000 HPLC system. The column was maintained at 35 °C. The analysis was performed using gradient solvents (A and B) containing 10 mM ammonium formate. Solvent A contained propanol: hexane:water (285:215:5, v/v/v) and solvent B contained propanol:hexane:water (285:215:40, v/v/v). All solvents were LC/MS grade. The column was eluted for 0–23 min with a linear gradient from 10 to 32 % B; 23–32 min using a linear gradient of 32–65% B; 32–35 min with a linear gradient of 65–100 % B; 35–62 min held at 100% B; 62–64 min with a linear gradient from 100 to 10 % B followed by an equilibration from 64 to 80 min at 10 % B. The instrument was operated with ESI probe in negative polarity mode. Analysis of LC/MS data was performed using software package Compound Discoverer (ThermoFisher Scientific) with an in-house generated analysis workflow and oxidized phospholipid database.

LC/ESI-MS analysis of TAGS was performed on a Dionex HPLC system (utilizing the Chromeleon software), consisting of a Dionex UltiMate 3000 mobile phase pump, equipped with an UltiMate 3000 degassing unit and UltiMate 3000 autosampler (sampler chamber temperature was set at 4 °C). The Dionex HPLC system was coupled to a hybrid quadrupole-orbitrap mass spectrometer, Q-Exactive (ThermoFisher) with the Xcalibur operating system. The instrument was operated in the positive ion mode (at a voltage differential of −5.0 kV, source temperature was maintained at 150 °C). TAG cations were formed through molecular ammonium adduction (+NH4). Analysis was performed at a resolution

of 140,000 for the full MS scan and 17,500 for the MS[2] scan in a data-dependent mode. The scan range for MS analysis was 300–1200 m/z with a maximum injection time of 128 ms using one microscan. A maximum injection time of 500 ms was used for MS[2] (high energy collisional dissociation (HCD)) analysis with collision energy set to 24. An isolation window of 1.0 Da was set for the MS and MS[2] scans. Capillary spray voltage was set at 4.5 kV, and capillary temperature was 320 °C. Sheath gas was set to eight arbitrary units and the S-lens Rf level was set to 60. TAGs were separated on a reverse phase column (Luna 3 μm C18 (2) 100 A, 150 × 1.0 mm, (Phenomenex)) at a flow rate of 0.065 ml/min. The column was maintained at 35 °C. The analysis was performed using gradient solvents (A and B) containing 0.1% NH4OH. Solvent A was methanol and solvent B was propanol. The column was eluted for 2 min from 0% B to 2% B (linear), from 3 to 6 min with a linear gradient from 2% solvent B to 3% solvent B, then isocratically from 3 to 18 min using 3% solvent B, 18 to 35 min with a linear gradient from 3% solvent B to 40% solvent B, 35 to 60 min using a linear gradient from 40 to 55% solvent B, then isocratically from 60 to 65 min at 55% solvent B then from 65 to 80 min from 55 to 0% B (linear) followed by equilibration from 80 to 90 min at 0% B. Standards for TAGs and their oxygenated metabolites were from Avanti Polar Lipids and Cayman Chemicals.

### Live Cell Imaging.

For the time-lapse imaging, BT-549 cells were seeded in a 6-well plate at 125,000 cells/well. After 24 h, stauporine (500 nM), αESA (25 μM), or ML162 (500 nM) were added along with 2 mM HEPES (pH 7.4) and then each well was overlaid with mineral oil and placed in a temperature-controlled imaging chamber (37 °C). Three fields in each well were imaged every 10 min for a 24 h period using a Nikon Eclipse TE300 inverted microscope equipped with a Nikon Plan Fluor 10x NA 0.30 objective and a QImaging Retiga EXi camera and using MetaVue software. Movies were assembled using the Fiji implementation of ImageJ[77]. For photomicrographs, cell images were taken using a Nikon Eclipse TE2000 inverted microscope equipped with Nikon Plan Fluor 10x NA 0.30 objective using Metavue (MetaMorph) or Ocular (QImaging) software.

### Detection of lipid peroxidation products.

The Click-iT Lipid Peroxidation Imaging kit (ThermoFisher Scientific) with Alexa Fluor 488 azide was used to detect macromolecules modified with breakdown products from peroxidized lipids according to the recommendations of the manufacturer. Cells were incubated with linoleamide alkyne (LAA) reagent for 16 h in experiments involving BSO, 4 h for experiments involving αESA, and 2 h in experiments using tung oil-conditioned medium before being fixed with 4% formaldehyde. DAPI was used to visualize DNA. Fluorescent micrographs were captured with a Nikon Eclipse TE2000 inverted microscope with a Nikon Plan Apo ×60 A/1.40 Oil using Metavue (MetaMorph) or Ocular (QImaging) software except in the experiment using tung oil-conditioned medium where images used for quantification were captured at ×20 magnification using a Zeiss Axio Observer inverted microscope and Zen Blue Software (Zeiss). The fluorescence signal intensity of individual cells was quantified using ImageJ (NIH).

### Small interfering RNA.

For siRNA knockdown experiments, cells were transfected (DharmaFECT 1, Horizon Discovery) with ON_TARGETplus SMART pools (Horizon Discovery) targeting individual *ACSL* family members, *GPX4* (Catalog ID: L-011676-00), *DGAT1* (Catalog ID: L-003922-00), *DGAT2* (Catalog ID: L-009333-00), *ATGL* (Catalog ID: L-009003-01) or a non-targeting siRNA pool (Pool #1, Catalog ID: D-001206-13, Horizon Discovery). For the *GPX4* siRNA experiment presented in Fig. 4e, a negative control targeting firefly luciferase was used (Luciferase GL2 Duplex, Catalog ID: D-001100-01, Horizon Discovery). For the *ATGL* siRNA experiments presented in Supplementary Fig. 6c–g, ON-TARGETplus a non-targeting control pool was used (Catalog ID: D-001810-10, Horizon Discovery). Efficiency of mRNA depletion was assessed 72 h post-transfection using qPCR or western blot in the case of ACSL1 (Cell Signaling, Antibody #4047, used 1:1000), GPX4 (Abcam, [EPNCIR144], ab125066, used 1:1000), DGAT1 (Abcam, [EPR13430-4] - N-terminal ab181180, used 1:5000), DGAT2 (ThermoFisher Scientific, PA5-103785, used 1:1000), and ATGL (Cell Signaling, Antibody #2138, used 1:1000). β-actin (Abcam, ab8227, used 1:5000) or GAPDH (Santa Cruz Biotechnology, sc-32233, used 1:5000) were used as loading controls. Secondary antibodies used were goat-anti-mouse-horseradish peroxidase (ThermoFisher Scientific, #31430, used 1:3000) and goat-anti-rabbit horseradish peroxidase (ThermoFisher Scientific, #31460, used 1:3000). In experiments to determine suppression of cell toxicity resulting from αESA or ML162 by silencing expression of individual *ACSL* genes, doses of αESA or ML162 were selected such that there was 5–15% remaining viability in cells transfected with non-targeting siRNA following 24 h of treatment.

### Lentivirus production.

Lentiviruses for transductions were packaged and collected from Lenti-X 293 T cells (Takara). The mammalian expression vector along with the lentivirus packaging vector (psPAX2) and envelope vector (pMD2.G) were co-transfected into Lenti-X 293 T using X-tremeGENE HP DNA transfection reagent (Roche). Lentiviruses were collected at 48, 72, and 96 h post-transfection and filtered using a 0.22 μm membrane. Target cells were incubated with virus-containing

medium and 2 μg/mL polybrene (Sigma Aldrich) for 24 h, and then allowed recovery for 24 h prior to selection for transductants.

**Stable cell line construction, CRISPR gene editing and gene over-expression**. CRISPR/Cas9 technology was used to generate the *ACSL1*-deficient lines. 20-bp guide RNA sequences were designed using the CRISPR Design Tool (http://crispr.mit.edu/), and expressed along with Cas9 using the plentiCRISPRv2 vector, which was a gift from Feng Zhang (Addgene plasmid # 52961; http://www.addgene.org/search/catalog/plasmids/?q=52961)[78]. Guide RNA sequences are as follows: Non-targeting, scrambled (5′-GCTAAGATCTCGACAACACT-3′), *ACSL1* (5′- GTTTC CGAGAGCCTAAACAA-3′). A second independent *ACSL1*-targeting guide was used to generate the line used in Supplementary Fig. 4g, h (5′- CAAGAGCCATC GCTTCAGCG-3′). After transduction with the relevant guide-containing plenti-CRISPRv2, and selection for cells resistant to puromycin (Invivogen, 2 μg/mL), single cell *ACSL1*-deficient clones were isolated by plating cells in 96-well plates at a concentration which 1 cell per 3 wells would be expected. Clonal populations were grown in RPMI-1640 containing 20% FBS without puromycin. Disruption of *ACSL1* was determined by western blot. Genomic sequencing confirmed distinct mutations.

Proteins were transgenically over-expressed using the pLX304 vector (Addgene plasmid #25890; http://www.addgene.org/search/catalog/plasmids/?q=25890). pLX304 was a gift from David Root[79]. For expression of *GPX4*, a construct containing *GPX4* cDNA (GenBank accession BC011836; CCSB Human ORFeome Clone Id 10338, Horizon Discovery), a V5 tag, and the 3′UTR of *GPX4* (GenBank accession BC011836) was generated using overlap extension PCR. The primers (5′ to 3′) used were as follows: *GPX4* ORF AttB1 F (GGGGACAAGTTTGTACAAA AAAGCAGGCTCAATGAGCCTCGGCCGGCCTTTGC), *GPX4* ORF OL R (ccacttt gtacaagaaagttgggtAGAAATAGTGGGGCAGGTCC), and *GPX4* ORF AttB1 R (GGGGACCACTTTGTACAAGAAAGCTGGGTTAATTTGTCTGTTTATTCCC ACAAGG). *ACSL1* cDNA was PCR amplified from the MGC Fully Sequenced Human *ACSL1* cDNA construct (GenBank accession BC050073; Clone Id 5267238, Horizon Discovery). The primers (5′ to 3′) used were as follows: *ACSL1* F AttB1 (GGGGACAAGTTTGTACAAAAAAGCAGGCTCAATGCAAGCCCATGAGCT GTTC) and *ACSL1* R AttB2 (GGGGACCACTTTGTACAAGAAAGCTGGGTT GTAAACCTTGATAGTGGAATAGAGG). To generate the *ACSL1* CRISPR-resistant mutant, silent mutations in each of the codons (seven total) in the *ACSL1* open reading frame recognized by the guide RNA were made using overlap extension PCR. PCR products were subcloned (BP reaction, ThermoFisher Scientific) into pDONR/Zeo (ThermoFisher Scientific) before Gateway cloning into the pLX304 lentiviral expression vector (LR reaction, ThermoFisher Scientific). In addition to the primers used to amplify the cDNA, the following primers (5′ to 3′) were used: *ACSL1* OL Primer 3 F (CATGCTTGGGTTCCCGCAAGCCAGACCAA CCCTATGAATG) and *ACSL1* OL Primer 2 R (GAATACAGGTGTCAAATAATG GCCCATGCTTGGGTTCCCGCAAGC). Following lentiviral transduction, blasticidin was used to select for cells containing the pLX304 expression vector (Invivogen, 8 μg/mL for BT-549 and 20 μg/mL MDA-MB-231). Proteins expressed from pLX304 were detected using an anti-V5 antibody (Cell Signaling, (D3H8Q) Rabbit mAb #13202, used 1:2000). eGFP was used as the negative control (cloned from Addgene plasmid #25899; http://www.addgene.org/search/catalog/plasmids/?q=25899)[79].

**Quantitative PCR**. Total RNA was isolated using an RNeasy kit (Qiagen) and tested for quality on a Bioanalyzer (Agilent Technologies). RNA concentrations were determined with a NanoDrop spectrophotometer (ThermoFisher Scientific). RNA was reverse transcribed using Moloney murine leukemia virus reverse transcriptase (Ambion- ThermoFisher Scientific) and a mixture of anchored oligo-dT and random decamers (Integrated DNA Technologies). Two reverse-transcription reactions were performed for each sample using either 100 or 25 ng of input RNA. Aliquots of the cDNA were used to measure the expression levels of the genes with the primers, and Power SYBR Green master mix (Applied Biosystems, Thermo-Fisher Scientific) on a 7900 HT sequence detection system (Applied Biosystems, ThermoFisher Scientific). Cycling conditions were 95 °C, 15 min, followed by 40 (two-step) cycles (95 °C, 15 s; 60 °C, 60 s). Ct (cycle threshold) values were converted to quantities (in arbitrary units) using a standard curve (four points, four-fold dilutions) established with a calibrator sample. The primers (5′ to 3′) used were as follows: ACSL1 (GACATTGGAAAATGGTTACCAAATG, GGCTCACT TCGCATGTAGATA), ACSL3 (CGAAGCTGCTATTTCAGCAAG, CTGTCACC AGACCAGTTTCA), ACSL4 (TCTTCTCCGCTTACACTCTCT, CTTATAAATT CTATCCATGATTTCCGGA), ACSL5 (GGAGAATACATTGCACCAGAGA, AC TCCTACTAAGGATGACCGT), and GPX4 (ACGTCAAATTCGATATGTTCA GC, AAGTTCCACTTGATGGCATTTC). 36B4 was used as the normalizer (CCC ATTCTATCATCAACGGGTACAA, CAGCAAGTGGGAAGGTGTAATCC).

**RNA-seq**. Total RNA was isolated using the RNeasy kit (Qiagen) from three independent replicates of MDA-MB-231 cells incubated for 5 h with ML162 (125 nM) or αESA (100 μM) and two replicates from the control incubated with the vehicle for αESA (methanol). Three biological replicate tung-oil or control MDA-MB-231 xenograft tumor samples were homogenized in ice-cold PBS using a dounce homogenizer and total RNA was isolated using the RNeasy kit (Qiagen).

The stranded mRNA-seq library was generated from 1000 ng of total RNA from each sample using Truseq stranded mRNA library kit (Illumina) according to the product instructions. In short, mRNAs were enriched twice via poly-T-based RNA purification beads, and subjected to fragmentation at 94 °C for 8 min via the divalent cation method. The first strand cDNA was synthesized by SuperscriptII (ThermoFisher Scientific) and random primers at 42 °C for 15 mins, followed by second strand synthesis at 16 °C for 1 h. During second strand synthesis, the dUTP was used to replace dTTP, thereby the second strand was quenched during amplification. A single 'A' nucleotide is added to the 3′ ends of the blunt fragments at 37 °C for 30 min. Adapters with illuminaP5, P7 sequences as well as indices were ligated to the cDNA fragment at 30 °C for 10 min. After Ampure bead (BD) purification (Beckman Coulter), a 15-cycle PCR reaction was used to enrich the fragments. PCR was set at 98 °C for 10 s, 60 °C for 30 s, and extended at 72 °C for 30 s. Libraries were again purified using AmPure beads, checked for quality check with a Bioanalyzer (Agilent Technologies) and quantified with Qubit (Invitrogen). Sample libraries were subsequently pooled and loaded to the HiSeq2500 and sequenced using a Hiseq rapid SRcluster kit and HiSeq rapid SBS kit (Illumina). Single 50 bp were generated for the bioinformatic analysis. Raw sequence reads were aligned to the human genome (hg38) using the Tophat algorithm[80] and Cufflinks algorithm[81] was implemented to assemble transcripts and estimate their abundance. Cuffdiff[82] was used to statistically assess expression changes in quantified genes in different conditions.

**Cell viability assays and reagents**. Cells were seeded in 96-well plates (Corning 3917, 3125–6250 cells per well) and treated with compounds 24 h after plating. Compounds were purchased from Cayman Chemical with the exception of stearic acid, oleic acid, adrenic acid, linolenic acid, linoleic acid, docosahexaenoic acid, conjugated linoleic acid (16413), bovinic acid, arachidonic acid, Z-VAD-FML, and (+)-α-tocopherol (T3634), which were purchased from Sigma Aldrich. RSL3 was purchased from Selleckchem, catalpic acid, α-calendic acid, β-calendic acid, and β-ESA were obtained from Larodan Fine Chemicals, and punicic acid was purchased from Matreya LLC. Initial aliquots of ferrostatin-1, deferoxamine, and ML162 were generous gifts of the Dixon Lab (Stanford) and additional quantities were purchased from Cayman Chemical. The inhibitor of FSP1[9] was purchased from ChemBridge Corp. Staurosporine was purchased from LC Sciences. Cell viability was measured using CellTiter-Glo Luminescent Cell Viability Assay (Promega) according to the manufacturer's instructions. Luminescence was measured on an EnSpire Alpha (Perkin Elmer) using the integrated software package. Data were normalized to vehicle-treated or sensitizing agent-alone controls and $IC_{50}$ curves were produced with GraphPad Prism (Version 8).

**Glutathione measurements**. Cellular glutathione was quantified using the GSH/GSSG-Glo kit (Promega) according to the instructions provided by the manufacturer. Drug-treated samples were normalized to parallel cell viability measurements using the CellTiter-Glo assay (Promega).

**Quantitation of GPX4 protein levels**. MDA-MB-231 were seeded at a density of 170,000 cells/well of a 6-well plate. After 24 h, Cells were incubated for 4 h with either vehicle (methanol or DMSO), ML162 (200 nM), RSL3 (200 nM), erastin (1 μM), or αESA (50 μM). Following treatments, cells were typsinized, pelleted, lysed with RIPA buffer. The western blot was performed using a GPX4 antibody (Abcam, [EPNCIR144], ab125066) (1:1000), β-actin antibody (Abcam, ab8227) (1:5000) for loading control, and goat anti-rabbit IgG (H + L) secondary antibody, HRP (ThermoFisher Scientific, 31460) (1:3000). ImageJ (NIH) was used for the densitometric quantification.

**Preparation of tung oil-conditioned medium**. Tung oil (Sigma, 440337)-conditioned medium was prepared by combining tung oil and RPMI-1640 complete medium in a 1:2 ratio and emulsifying the components by rotating for 45 min at room temperature. The emulsion was then centrifuged at 2000 × g for 10 min at room temperature. The oil phase was removed and discarded. The aqueous phase was filtered and frozen until further use.

**Mouse xenografts**. This study was reviewed and approved by the Fox Chase Cancer Center Institutional Animal Care and Use Committee (Protocol #16-10) and complies with ethical regulations for animal research. Mice were housed in a dedicated laboratory animal facility with 12-h light:dark cycle, at 70 F +/−2 degrees, and 40–70% relative humidity. Orthotopic xenografts were generated by implanting 2.5 million MDA-MB-231 cells in 100 μL phosphate-buffered saline (PBS) mixed with 100 μL growth factor-reduced Matrigel (Corning) bilaterally into the fourth inguinal fat pad of four- to six-week-old female NOD.Cg-*Prkdc^scid^Il2rg^tm1Wjl^*/SzJ (NSG) mice. After 14 days, when tumors were roughly 50 mm³, animals were randomized into treatment groups. Mice were treated on weekdays with either safflower oil (Whole Foods 365) and Tung oil (Sigma-Aldrich 440337) at a dose of 100 μL administered by oral gavage. L-BSO (Sigma) was administered via the drinking water (20 mM) ad libitum, as previously reported[23,83].

**Tissue preparation, histology, and immunohistochemistry**. Tumor sections and lung sections were fixed in 10% formalin for 24–48 h, dehydrated, and embedded in paraffin. Immunohistochemical staining was performed by standard methods. To quantify lung metastases, lungs from three mice per treatment group were fixed, sectioned and hematoxylin and eosin stained. Ten equally sized regions (2 from each lobe) were randomly selected and scored manually for areas occupied by cancerous tissue. Micrographs were captured with a Nikon DS-Fi1 camera (Melville, NY, USA) on a Nikon Eclipse 50i microscope and quantified using a ScanScope CS5 slide scanner (Aperio).

## Data availability

All data generated or analyzed during this study are included in this published article (and its supplementary information files) or are available from the corresponding author on reasonable request. Lipidomic data is available at the NIH Common Fund's National Metabolomics Data Repository (NMDR) website, the Metabolomics Workbench, https://www.metabolomicsworkbench.org where it has been assigned Project ID PR000997. The data can be accessed directly via its Project https://doi.org/10.21228/M8TH75 [https://www.metabolomicsworkbench.org/data/DRCCMetadata.php?Mode=Project&ProjectID=PR000997]. RNAseq data is available from the National Center for Biotechnology GEO database (GSE162069). Source data are provided with this paper.

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

## Acknowledgements

This work was supported in part by the Office of the Assistant Secretary of Defense for Health Affairs through the Breast Cancer Research Program of the CDMRP under Award No. W81XWH-19-1-0481. Additional support was provided by National Institutes of Health Awards GM083025, U19AI068021, HL114453-06, CA165065-06, NS076511, the PA Breast Cancer Coalition, the Fifth District AHEPA Cancer Research Foundation, the Spurlino Family Foundation, the Rita Hollman Foundation, the Eileen Stein Jacoby Fund, and Translational Research and In Vino Vita awards from Fox Chase Cancer Center. Supported in part by Award Number P30 CA006927 from the National Cancer Institute of the National Institutes of Health. We thank Drs. S. Balachandran and J. Karanicolas for comments on the manuscript.

## Author contributions

Experiments and data analysis were carried out by A.B., T.S., Y.Y.T., V.A.T., S.S., E.N., K.M., Y.Z., K.Q.C., Y.T., and J.R.P.. S.D. and M.C. provided reagents. A.S. conducted the PRISM cell profiling study. A.B., H.B., V.E.K., U.R., and J.R.P. supervised the work. A.B. and J.R.P. wrote the manuscript with input from the co-authors.

## Competing interests

Ulrike Rennefahrt is a former employee of Metanomics Health, GmbH. Marcus Conrad is an inventor on patents for some of the compounds described herein, and shareholder of ROSCUE Therapeutics GmbH. The remaining authors declare no competing interests.
