## [Peer Review File · Nature Communications]

REVIEWER COMMENTS

Reviewer #1 (Remarks to the Author):

The authors have significantly revised the manuscript and addressed weaknesses of the previous submission. This new manuscript is much improved, interesting, and important.

A major remaining question is what to make of the observed enrichment of alpha-ESA in DAGs and TAGs, and yet the preferential oxidation of PEs and CLs. Is the model that alpha-ESA must transit through the neutral lipid pool before reaching membrane phospholipids? If so, why would this be necessary? Do other PUFAs not transit through this route to phospholipids, and is this why these are other PUFAs are not able to induce ferroptosis alone?

It is argued on line 347 that deleterious lipid oxidation may initiate in lipid droplets, but the experimental analysis revealed no oxidation of PUFA-containing TAGs, so this statement appears contradictory. Moreover, I would also question the value of suggesting that (line 351) "...triggering ferroptosis may results in the liberation of alpha-ESA from TAGs...", since one of the main take home messages of this paper is that alpha-ESA alone is sufficient to trigger ferroptosis alone. This single agent activity is the phenomenon which requires a clear explanation and model, not the case of sensitization to other inducers of ferroptosis or liberation of alpha-ESA as a secondary/downstream consequence of ferroptosis induction. Some clarification on these points (not new experiments) would enrich the Discussion.

The Discussion might also benefit from some consideration of the alpha-ESA SAR and what this implies. It might, for example, imply that to promote ferroptosis the PUFA must be a substrate of a structure-specific enzyme that can modify it.

Typo is line 253. Extra 'S'. "DAGs/TAGSs,"

Reviewer #2 (Remarks to the Author):

I appreciate the lack of direct molecular marker of ferroptosis. However, it is long known that PUFA can induce toxicity due to disruption of membranes. There are both in cellular and ex cellular studies investing the mechanism of this. An important alternative mode of action to what the authors suggest is that cell death upon addition of PUFA can simply be due to membrane disruption. The fact that there is death when GSH depleted (which can be rescued by adding antioxidants) and it is not zVAD dependent does not make it ferroptosis.

1) The concept of investigating the cell viability by addition of different fatty acids is not a novel concept. This is has been explored in many studies, including the distribution of these fatty acids to

different lipid pools. it is know that unsaturated fatty acid add backs have a higher propensity to induce cell death and they are incorporated to phospholipids and glycerolipids. Hence, the results presented in “alphaESA is incorporated into diverse lipid pools” section are completely expected. The authors should cite the relevant studies in order to prevent confusion on the novelty of these findings.

2) the “structure activity relationship” (toxicity) with different fatty acids should be taken with a grain of salt as the solubility of these fatty acids vary substantially by the degree of unsaturation. Furthermore, their uptake depend on the different cell surface receptors that might have different expression levels.

3) the contribution of ACSL1: The data presented does not support that ACSL1 is the only isoform that mediates the incorporation of alphaESA to different lipids. This is also indicated by the authors (page 10 lines 247-248) . Revision in the text are recommended to clarify that ACSL1 “in part” mediates processing of alphaESA.

4) Viability data with zVAD should be added to Fig1.

5) only one nontransformed cell line is tested. To suggest a cancer specific mode of action, more nontransformed cell lines should be added.

6) The contribution of TAGs to alphaESA toxicity is contradictory to the other findings in this paper. TAGs are not membrane localized, as such the inhibitor of their synthesis should not cause membrane damage.

7) Related to this, I find the involvement of lipid droplets in potentiating lipid oxidation is somewhat controversial and not back up by enough data. There are many studies supporting a “protective” role for these organelles (not just reference 49). I suggest to tone down this section in discussion especially in the absence of any mass spec data on oxidized TAGs.

REVIEWER COMMENTS AND RESPONSES

We thank the reviewers for their careful review of our revised manuscript. Their constructive comments led to numerous improvements in our manuscript including 6 new figure panels (Figs. 5g, 5h, 6d-g) and 1 revised figure panel (Fig. 1b), 9 new supplementary figure panels (Figs. S2d, S5i, S6c-g, S6i, S6j), 1 revised supplementary figure panel (Fig. S2f), 1 revised supplementary table (Table 6) and a more complete mechanistic model, as outlined below in the point-by-point responses. Textual changes are highlighted using “track changes” in the Word document.

Reviewer #1 (Remarks to the Author):

The authors have significantly revised the manuscript and addressed weaknesses of the previous submission. This new manuscript is much improved, interesting, and important.

A major remaining question is what to make of the observed enrichment of alpha-ESA in DAGs and TAGs, and yet the preferential oxidation of PEs and CLs. Is the model that alpha-ESA must transit through the neutral lipid pool before reaching membrane phospholipids? If so, why would this be necessary? Do other PUFAs not transit through this route to phospholipids, and is this why these other PUFAs are not able to induce ferroptosis alone?

It is argued on line 347 that deleterious lipid oxidation may initiate in lipid droplets, but the experimental analysis revealed no oxidation of PUFA-containing TAGs, so this statement appears contradictory.

The reviewer put his/her finger on an important remaining question that we have now addressed. Based on support from new data and prior literature, we now propose a more complete model for the role of the neutral lipid pool in ESA-induced ferroptosis. A key feature of our model is that ESA is locally concentrated in neutral lipids. Our lipidomic analysis demonstrates that TAG lipids increase by > 2-fold in cells treated with ESA for only 3 hours. At this timepoint, >70% of TAGs already contain at least one 18:3 acyl chain. By contrast, no phospholipid class has more than 20% 18:3-containing acyl chains. In addition, a new paper published this month (Do et al. *J Org Chem*, 2020) reported that conjugated linolenic acid undergoes free radical oxidation propagation over 8-fold faster than the corresponding non-conjugated fatty acid, which, together with local concentration in neutral lipids, provides a potential chemical basis for the differential ability of these lipids to induce ferroptosis.

A more comprehensive analysis of our mass spectrometry lipidomic data now enabled us to identify two oxidized TAG species that are increased upon ESA treatment in an ACSL1-dependent manner (Fig 5g,h). Unlike non-conjugated acyl chains, ESA as well as ESA-containing TAGs are known to undergo oxidative polymerization to form high molecular weight species. Polymerization rate is directly related to monomer concentration, providing an explanation for how local concentration can promote this reaction. Because our mass spectrometry analysis was limited to species <1,200 molecular weight, high molecular polymer products were not quantified, potentially leading us to underestimate TAG oxidation. However, we did identify the formation of adducts of ESA derivatives with diacylglycerol containing conjugated acyl chains (Figs. 5g-h), supporting the possibility that larger covalent polymers also form.

Finally, we used tung oil, which is comprised of TAGs with ~80% esterified ESA, as a model for ESA-rich neutral lipids in order to address the question of how oxidation of ESA in the lipid droplets could mediate phospholipid oxygenation. We created an emulsion of tung oil and cell culture medium in the presence of oxygen, separated the phases, and used the aqueous phase to culture TNBC cells. This tung oil-conditioned cell culture medium resulted in cellular lipid peroxidation and cell death consistent with ferroptosis (Fig 6f, g). These data suggest that tung oil releases a water-soluble “mediator” (whose identity is still unknown) that can trigger ferroptosis when added to cells. These new results are consistent with a model where oxidative reactions in neutral lipids could release a diffusible signal responsible for triggering subsequent peroxidation of membrane phospholipids. Notably, such hydrophilic intermediates would have greater access to phospholipids present in planar membranes compared to neutral lipids in spherical lipid droplets, which could also contribute to why we observe primarily oxidation products of membrane lipids rather than neutral lipids.

Thus, our overall model postulates that ACSL1-mediated ESA accumulation in TAGs leads to oxidative polymerization and release of reactive mediators that propagate lipid oxidation in lipid bilayers. We believe this new model, supported by multiple new figure panels, provides a much clearer mechanism for ESA-triggered ferroptosis.

I would also question the value of suggesting that (line 351) “...triggering ferroptosis may results in the liberation of alpha-ESA from TAGs...”, since one of the main take home messages of this paper is that alpha-ESA alone is sufficient to trigger ferroptosis alone. This single agent activity is the phenomenon which requires a clear explanation and model, not the case of sensitization to other inducers of ferroptosis or liberation of alpha-ESA as a secondary/downstream consequence of ferroptosis induction. Some clarification on these points (not new experiments) would enrich the Discussion.

We agree and have deleted the discussion of liberation of ESA from TAGs as discussed above. Furthermore, we recently found that our prior data showing that the ATGL inhibitor atglistatin suppressed ESA-induced ferroptosis was not phenocopied by ATGL siRNA knockdown (new Supplementary Fig S6d-j), suggesting that release of fatty acids from neutral lipids is not a required step and that atglistatin may instead be acting as a radical trap.

The Discussion might also benefit from some consideration of the alpha-ESA SAR and what this implies. It might, for example, imply that to promote ferroptosis the PUFA must be a substrate of a structure-specific enzyme that can modify it.

This point was raised by both reviewers. As reviewer 1 suggests, the differential potency of the fatty acids tested could reflect differences in their ability to be metabolized by specific enzymes that contribute to regulating ferroptosis. However, as reviewer 2 points out, we cannot exclude the possibility that potency differences instead reflect their physical properties, such as solubility.

Consequently, we have refrained from drawing a strong conclusion from the SAR. In the revised text (line 184-186) we now state: “Whether

the differential potency of these related molecules is related to their metabolism by specific enzymes or due to differences in solubility or other physicochemical properties is an important open question.”

Typo is line 253. Extra 'S'. "DAGs/TAGSs,"
Corrected. Thank you.

Reviewer #2 (Remarks to the Author):

I appreciate the lack of direct molecular marker of ferroptosis. However, it is long known that PUFA can induce toxicity due to disruption of membranes. There are both in cellular and ex cellular studies investigating the mechanism of this. An important alternative mode of action to what the authors suggest is that cell death upon addition of PUFA can simply be due to membrane disruption.

While various fatty acids have been previously shown to promote cell death, here we define a distinct cell death process, ferroptosis, that is specifically associated with PUFAs containing a conjugated triene system. In new data (Supplementary Figure S2f) we highlight this fact by characterizing cell death caused by the non-conjugated PUFA arachidonic acid. The modest death induced by arachidonic acid cannot be rescued by fer-1, liproxstatin, or deferoxamine, thus differentiating cell death caused by a non-conjugated PUFA from ESA-induced cell death. Furthermore, the ability of antioxidants to suppress ESA-induced death argues that ESA incorporation is not simply due to physical membrane disruption.

The fact that there is death when GSH depleted (which can be rescued by adding antioxidants) and it is not zVAD dependent does not make it ferroptosis.

We have toned down our claims about BSO inducing ferroptosis. For example:

“Thus, glutathione depletion triggers death consistent with ferroptosis in BT-549 cells and identifies lipid hydroperoxides as the lethal ROS species underlying glutathione addiction in these cells.” Line 84

“Fer-1 suppressed BSO toxicity in half of the triple-negative breast cancer (TNBC) cell lines tested, consistent with ferroptosis.” Line 88

1) The concept of investigating the cell viability by addition of different fatty acids is not a novel concept. This is has been explored in many studies, including the distribution of these fatty acids to different lipid pools. it is know that unsaturated fatty acid add backs have a higher propensity to induce cell death and they are incorporated to phospholipids and glycerolipids. Hence, the results presented in “alphaESA is incorporated into diverse lipid pools” section are completely expected. The authors should cite the relevant studies in order to prevent confusion on the novelty of these findings.

While we agree that the observation that ESA is incorporated into diverse lipid pools is, on its own, not unexpected, the precise lipid species that incorporate ESA have not been previously reported. Furthermore, an emerging theme in lipid research is that different fatty acids can induce different types of cell death via different mechanisms. For example, while saturated fatty acids such as palmitate induce apoptosis, we demonstrate that conjugated linolenic fatty acids induce ferroptosis. Our discovery that ESA becomes enriched in neutral lipid pools, though perhaps not surprising on its own, led to our discovery of the critical role of this lipid pool in ESA-induced ferroptosis. In addition, it provided a mechanistic explanation for the role of ACSL1 in this process, since ACSL1 promotes the incorporation of ESA preferentially into this lipid pool. Thus our lipidomic analysis of ESA-treated cells was a key enabling finding to understand this unusual mechanism of cell killing.

2) the “structure activity relationship” (toxicity) with different fatty acids should be taken with a grain of salt as the solubility of these fatty acids vary substantially by the degree of unsaturation. Furthermore, their uptake depend on the different cell surface receptors that might have different expression levels.

Please see the response to the related comment from reviewer 1.

3) the contribution of ACSL1: The data presented does not support that ACSL1 is the only isoform that mediates the incorporation of alphaESA to different lipids. This is also indicated by the authors (page 10 lines 247-248) . Revision in the text are recommended to clarify that ACSL1 “in part” mediates processing of alphaESA.

We agree and have revised the text (lines 249-250) to read “alphaESA toxicity, however, was significantly suppressed by knockdown of ACSL1, though other ACSL isoforms may contribute.”

4) Viability data with zVAD should be added to Fig1.

The requested data has been added in updated panel Fig. 1b.

5) Only one non-transformed cell line is tested. To suggest a cancer specific mode of action, more nontransformed cell lines should be added.

We have examined the sensitivity of an additional non-transformed mammary cell line, MCF-12A, and confirmed resistance to ESA (new Supplementary Fig S2d), in line with our data for MCF-10A (Fig. 2e). Nevertheless, the lack of very many non-transformed mammary cell lines limits the strength of our suggestions of a cancer-specific mode of action and we have eliminated any statements that such is the case.

6) The contribution of TAGs to alphaESA toxicity is contradictory to the other findings in this paper. TAGs are not membrane localized, as such the inhibitor of their synthesis should not cause membrane damage.

See response to the related comment of Reviewer 1.

7) Related to this, I find the involvement of lipid droplets in potentiating lipid oxidation is somewhat controversial and not back up by enough data. There are many studies supporting a “protective” role for these organelles (not just reference 49). I suggest to tone down this section in discussion especially in the absence of any mass spec data on oxidized TAGs.

We agree that our data implicating neutral lipid droplets in ESA-induced ferroptosis is novel and will be a focus of future studies. We have provided a more complete description of a model in a revised Discussion section based on support from new data and prior literature. As described in our response to Reviewer 1, a more thorough analysis of our mass spectrometry lipidomic data enabled us to identify two oxidized TAG species that were increased with α ESA treatment in an ACSL1-dependent manner (Fig 5g,h). Moreover, we speculate that our analysis of low molecular weight TAG species may underestimate TAG oxidation that is well known to be associated with the tendency of conjugated fatty acids to undergo oxidative polymerization. Because our mass spectrometry analysis was limited to species <1,200 molecular weight, these oxidized, polymeric reaction products were not quantified, potentially leading us to underestimate TAG oxidation. The hypothesis is also supported by our identification of adducts of ESA moieties with diacylglycerol containing esterified ESA (Figs. 5g-h). Furthermore, as outlined in the response to Reviewer 1, we also present new data (Fig 6f, g) supporting the possibility that diffusible mediators derived from ESA-containing TAGs can propagate lipid peroxidation, potentially to membrane lipids.

Finally, we have added an additional reference to a prior study (Jarc et al. 2018) demonstrating that lipid droplets play a protective role from lipid peroxidation in other contexts. Thus, our work highlights that this normally protective mechanism may actually promote oxidation of conjugated fatty acids by concentrating them together in neutral lipid pools. We think this provides an intriguing and more nuanced perspective on the role of lipid droplets.

REVIEWERS' COMMENTS

Reviewer #1 (Remarks to the Author):

The authors have addressed my comments and this interesting study should be published.

Scott J. Dixon

Reviewer #2 (Remarks to the Author):

The revised version addresses my concerns and suggestions. Thank you.